JCB Journal of Cell Biology

# Binucleated human hepatocytes arise through late cytokinetic regression during endomitosis M phase

Gabriella S. Darmasaputra[1], Cindy C. Geerlings[1], Susana M. Chuva de Sousa Lopes[2], Hans Clevers[1,3], and Matilde Galli[1]

Binucleated polyploid cells are common in many animal tissues, where they arise by endomitosis, a non-canonical cell cycle in which cells enter M phase but do not undergo cytokinesis. Different steps of cytokinesis have been shown to be inhibited during endomitosis M phase in rodents, but it is currently unknown how human cells undergo endomitosis. In this study, we use fetal-derived human hepatocyte organoids (Hep-Orgs) to investigate how human hepatocytes initiate and execute endomitosis. We find that cells in endomitosis M phase have normal mitotic timings, but lose membrane anchorage to the midbody during cytokinesis, which is associated with the loss of four cortical anchoring proteins, RacGAP1, Anillin, SEPT9, and citron kinase (CIT-K). Moreover, reduction of WNT activity increases the percentage of binucleated cells in Hep-Orgs, an effect that is dependent on the atypical E2F proteins, E2F7 and E2F8. Together, we have elucidated how hepatocytes undergo endomitosis in human Hep-Orgs, providing new insights into the mechanisms of endomitosis in mammals.

## Introduction

Polyploid cells, which contain more than two pairs of homologous chromosomes, are found in many tissues of diploid species, including mammals. Somatic polyploidization occurs during defined moments in development and is thought to be crucial for increases in metabolic output (Rios et al., 2016; Rossant and Cross, 2001), cell and organism size (Flemming et al., 2000), and to maintain specialized cell functions (Unhavaithaya and Orr-Weaver, 2012; Trakala et al., 2015; Wang et al., 2018). Polyploidization has also been shown to have a protective role against genotoxic stress by buffering the effect of detrimental genetic aberrations (Zhang et al., 2018; Mehrotra et al., 2008; Sladky et al., 2020). In humans, polyploid cells arise in tissues such as the liver, pancreas, mammary glands, placenta, and bone marrow (Pandit et al., 2013; Edgar et al., 2014; Orr-Weaver, 2015). In these tissues, cells become polyploid by undergoing non-canonical cell cycles, so-called endocycles, in which they replicate their DNA but do not divide. Two types of endocycles have been described that give rise to polyploidy: endoreplication and endomitosis. In endoreplication, cells alternate between S and G phases, duplicating their genomic DNA without ever entering M phase. In endomitosis, cells enter M phase but exit prematurely before completing cell division. Depending on the timing of M phase exit, daughter cells can either be mononucleated or binucleated.

An outstanding question in cell-cycle research is how cells transition from canonical to non-canonical cell cycles. Most prior work has focused on the transition from canonical cycles to endoreplication, which has been extensively studied in plants and insects and relies on downregulation of M-CDK, the kinase complex that drives M phase (Edgar et al., 2014). By suppressing M-CDK activity, cells cannot enter M phase, and this is sufficient to trigger endoreplicative cycles in animals and yeast (Hayles et al., 1994; Hayashi, 1996; Trakala et al., 2015; van Rijnberk et al., 2022). On the other hand, less is known about how cells transition to endomitosis cycles. In endomitosis, M-CDK needs to become active to initiate M phase, yet cytokinesis needs to be inhibited to block cell division. The mechanisms by which endomitotic cells inhibit cytokinesis differ per cell type and even during different stages of development of the same cell type. For example, during M phase of the first endomitosis cycle of megakaryocytes, cells undergo cleavage furrowing followed by regression of the furrow, whereas there is no cytokinetic ingression in subsequent endomitosis cycles (Geddis et al., 2007; Gao et al., 2012). Also in other endomitotic cells, such as mouse cardiomyocytes (Engel et al., 2006; Leone et al., 2018), hepatocytes (Guidotti et al., 2003; Celton-Morizur et al., 2009; Margall-Ducos et al., 2007; Pandit et al., 2012), and mammary cells (Rios et al., 2016), cell division is inhibited at different stages of M phase. Despite different mechanisms to inhibit cell division between different endomitotic cell types, downregulation or inhibition of key cytokinesis regulators seems to be a common feature.

[1]Hubrecht Institute, Royal Netherlands Academy of Arts and Sciences, University Medical Center Utrecht, Utrecht, Netherlands;   [2]Department of Anatomy and Embryology, Leiden University Medical Center, Leiden, Netherlands;   [3]Oncode Institute, Utrecht, Netherlands.

Correspondence to Matilde Galli: m.galli@hubrecht.eu.

Although most polyploid cells are considered terminally differentiated and post-mitotic, a few types of polyploid cells have been found to remain proliferative. In *Drosophila*, rectal papillae cells become polyploid through endoreplication but can switch back to mitotic divisions during later development or upon injury (Fox et al., 2010; Schoenfelder et al., 2014). Similarly, in damaged mouse livers, polyploid hepatocytes are able to divide and contribute to the regenerating tissue (Duncan et al., 2010; Matsumoto et al., 2020; Chen et al., 2020). The ability of cells to switch back from non-canonical cycles to canonical cycles suggests that the inhibition of cell division is reversible. This phenomenon is likely important for the regenerative capacity of the liver as it has been shown that polyploid hepatocytes contribute to liver regeneration in mice (Miyaoka et al., 2012; Chen et al., 2020; Matsumoto et al., 2020). Thus, deciphering the molecular mechanisms by which hepatocytes switch between canonical and non-canonical cycles will allow a deeper understanding of how tissues modulate the percentages of polyploid cells and may identify novel factors that control regeneration.

The mammalian liver is an attractive model to study the regulation of non-canonical cell cycles, as mature hepatocytes have been reported to perform both canonical and endomitotic cell cycles throughout their lifetime. In humans, hepatocytes range in ploidy from 2N to 8N, and up to 20% are binucleated (Gahan and Middleton, 1984; Kudryavtsev et al., 1993; Knouse et al., 2014; Heinke et al., 2022). In a healthy liver, <0.01% of hepatocytes are actively cycling (Delhaye et al., 1996; Walesky et al., 2020). However, upon injury, hepatocytes are able to proliferate and regrow the organ to an equivalent of the original size. Proliferating mouse hepatocytes require Wnt/β-catenin activity (Planas-Paz et al., 2016), but high Wnt/β-catenin activity by itself does not induce proliferation (Sun et al., 2020; Planas-Paz et al., 2016). This is in contrast with many other mammalian epithelia where the degree of proliferation strongly correlates with Wnt activity (Kretzschmar and Clevers, 2017), suggesting that the Wnt/β-catenin axis may have an unconventional role in controlling hepatocyte cell cycles.

Hepatocyte endomitosis has thus far mostly been studied in rodents, and it is currently unknown how human hepatocytes undergo endomitosis. In mouse livers, hepatocytes undergo canonical cell cycles during embryogenesis but transition to endomitosis cycles during postnatal development (Brodsky and Uryvaeva, 1985). Here, the activity of the atypical E2F transcription factors E2F7 and E2F8 drives endomitosis cycles by repressing cytokinesis genes (Pandit et al., 2012). It is unknown whether WNT/β-catenin signaling and E2F7/8 regulate endomitosis in human hepatocytes. Investigation of hepatocyte canonical and non-canonical cycles in vitro is not trivial, as primary hepatocyte cultures are limited in their proliferative capacity (Levy et al., 2015) and require a three-dimensional structure for proper polarization and proliferation (Zeigerer et al., 2017; Lorenz et al., 2018). Immortalized non-cancerous hepatocyte cell lines are also unsuitable for cell-cycle analyses as they often have impaired physiology (Sefried et al., 2018). In this study, we used fetal tissue–derived human hepatocyte organoids (Hep-Orgs) (Hu et al., 2018) as a model to study the regulation of endomitosis. Hep-Orgs can be cultured long-term

and be genetically modified, allowing integration of fluorescently tagged markers and live imaging of canonical and endomitosis M phases (Artegiani et al., 2020). We find that human hepatocytes undergoing endomitosis initiate cytokinesis with normal timing and morphology but regress their cytokinetic furrows during late M phase. Our immunofluorescence analyses of cells undergoing regression suggest that endomitotic cells have defects in tethering the midbody to the cell cortex. We also find that inhibition of WNT signaling increases the percentage of binucleated cells, a process that is dependent on E2F7 and E2F8. Together, our work reveals a novel mechanism for the inhibition of cytokinesis during endomitosis M phase and provides support for an evolutionary conserved function of WNT and E2F7/E2F8 in the transition to non-canonical cell cycles.

## Results

### Hep-Orgs represent a model to study canonical and endomitosis M phases in vitro

To determine whether Hep-Orgs can be used as a model to study the regulation of endomitosis, we first determined the percentage of binucleated cells in two Hep-Org lines derived from different donors, as well as a cholangiocyte organoid (Chol-Org) line, which was also derived from human liver but is composed of cholangiocytes rather than hepatocytes. We generated organoid strains expressing a nuclear GFP-NLS marker and used CellMask Orange dye to mark the cell membrane (Fig. 1 A). Consistent with published observations on the frequencies of binucleated hepatocytes in fetal human hepatocytes (Kudryavtsev et al., 1993), we observed an average of 5 and 15% binucleated cells in two independent Hep-Org lines (Fig. 1 B). In contrast, binucleated cells are rarely found (0.8%) in Chol-Orgs (Fig. 1 B). The percentage of binucleated cells in each organoid ranged between 0 and 32% and did not show significant differences between organoids of different sizes (Fig. 1 C).

To visualize endomitosis in Hep-Orgs, we performed long-term live imaging of Hep-Orgs expressing a nuclear GFP marker and endogenously tagged E-cadherin-tdTomato to mark cell membranes (referred to as GFP-NLS/E-cadherin-tdTomato), or endogenously tagged β-tubulin-mNeonGreen to visualize the mitotic spindle (referred to as Tubulin-mNeon) (Fig. 1, D and F). Cells undergoing M phase were identified based on either the loss of a confined GFP-NLS signal during nuclear envelope breakdown (NEB) in the GFP-NLS/E-cadherin-tdTomato line or by the appearance of spindle poles in the Tubulin-mNeon line (Fig. 1, D and F; and Fig. 2 A). We categorized different types of M phases depending on the number of nuclei in the mother and daughter cells. We found that 90% of M phases in both Hep-Org lines were canonical, where a mononucleated mother cell gives rise to two mononucleated daughter cells (Fig. 1 D, top panel). In contrast, 5–8% of M phases showed a mononucleated mother cell giving rise to one binucleated daughter cell, which we classify as endomitosis M phases (Fig. 1 D, bottom panel).

Finally, we also observed binucleated cells entering M phase (Fig. 1 F), showing that polyploid cells in Hep-Org

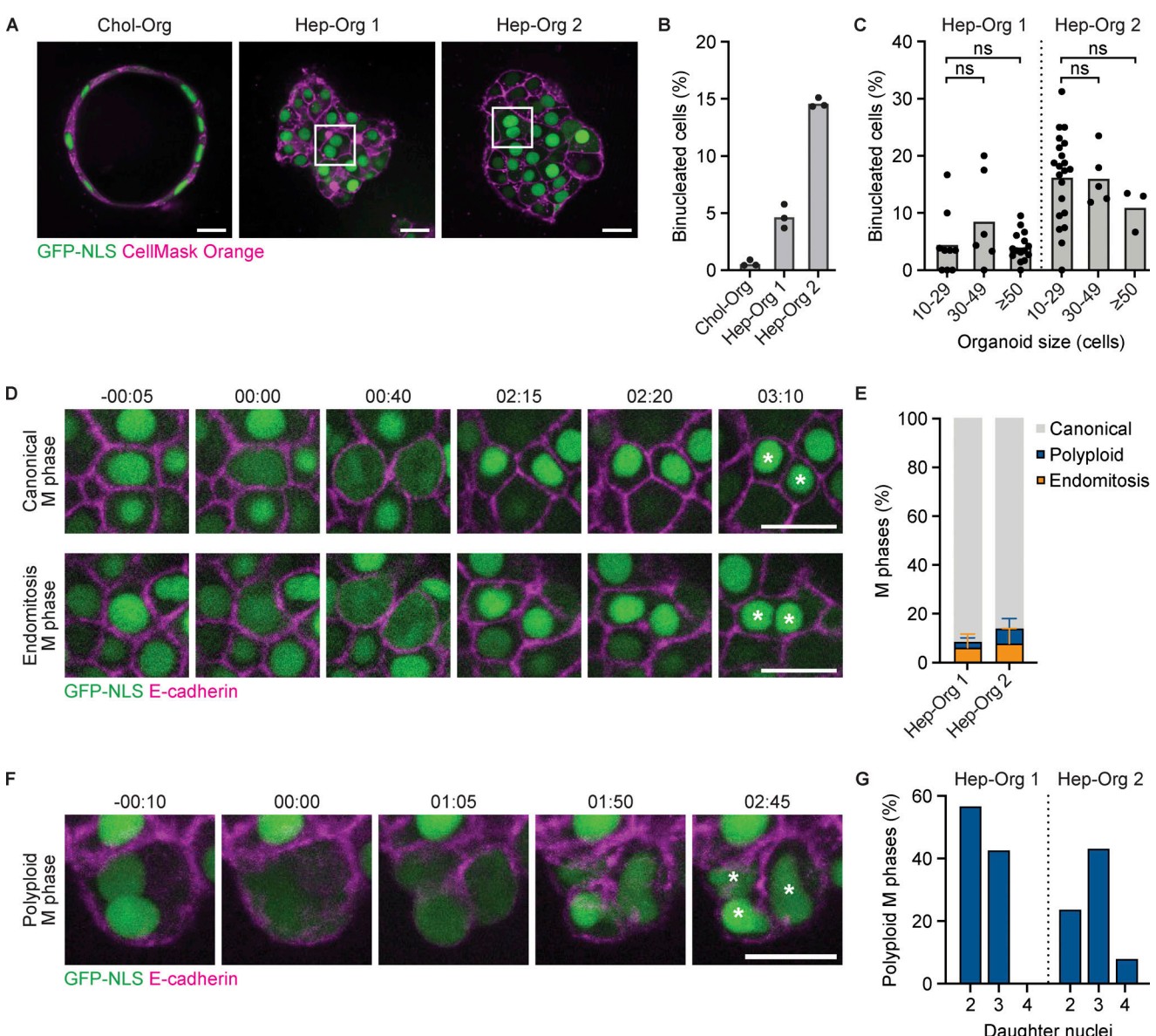

Figure 1. **Human hepatocytes undergo canonical, endomitosis, and polyploid M phases in Hep-Orgs. (A)** Representative images of Chol-Org and Hep-Org lines expressing GFP-NLS (green) and stained with CellMask Orange (magenta) to mark nuclei and membranes, respectively. Images show one plane in the center of each organoid. Examples of binucleated cells are indicated in the white squares. **(B)** Percentage of binucleated cells in Chol-Org and Hep-Org lines. Columns depict mean percentages ($N$ = 3 experiments, 200–300 cells analyzed per experiment). **(C)** Percentage of binucleated cells per organoid plotted against organoid size ($N$ = 3 experiments, 20–30 organoids analyzed per Hep-Org line). Each dot represents one organoid, with the mean depicted in column (ns = not significant, Student's $t$ test, two-tailed). **(D)** Stills from live imaging of GFP-NLS/E-cadherin-tdTomato Hep-Org 1 line showing canonical (top) and endomitosis (bottom) M phase. Time is relative to NEB in h:min. White asterisks mark daughter cells. **(E)** Percentage of types of M phases observed during live-imaging experiments of GFP-NLS/E-cad-tdTomato Hep-Org 1 line and Tubulin-mNeon Hep-Org 2 line ($N$ = 5 experiments, >175 events analyzed per line). Error bars represent standard deviation. **(F)** Stills from live imaging of GFP-NLS/E-cadherin-tdTomato Hep-Org 1 line showing polyploid M phase. Time is relative to NEB in h:min. White asterisks mark daughter cells. **(G)** Percentage of polyploid M phases segregating their DNA content to two, three, or four daughter nuclei ($N$ = 5 experiments, >7 events per line). Scale bars in A, D, and F represent 50 µm.

continue cycling, as has been shown previously (Artegiani et al., 2020). These divisions, which we refer to as polyploid M phases, give rise to either two, three, or four daughter nuclei (Fig. 1 G). Taken together, we observe canonical, endomitosis, and polyploid M phases in Hep-Orgs, rendering them an attractive model to study the transition between canonical and non-canonical cell cycles in human hepatocytes in vitro.

## Cells undergoing endomitosis M phase have normal mitotic timings but regress their cytokinetic furrow during late cytokinesis

Previous studies on endomitosis in mouse megakaryocytes, hepatocytes, and cardiomyocytes have identified distinct mechanisms to inhibit cytokinesis in endomitosis M phase (Geddis et al., 2007; Gao et al., 2012; Margall-Ducos et al., 2007; Celton-Morizur et al., 2009; Engel et al., 2006; Leone et al.,

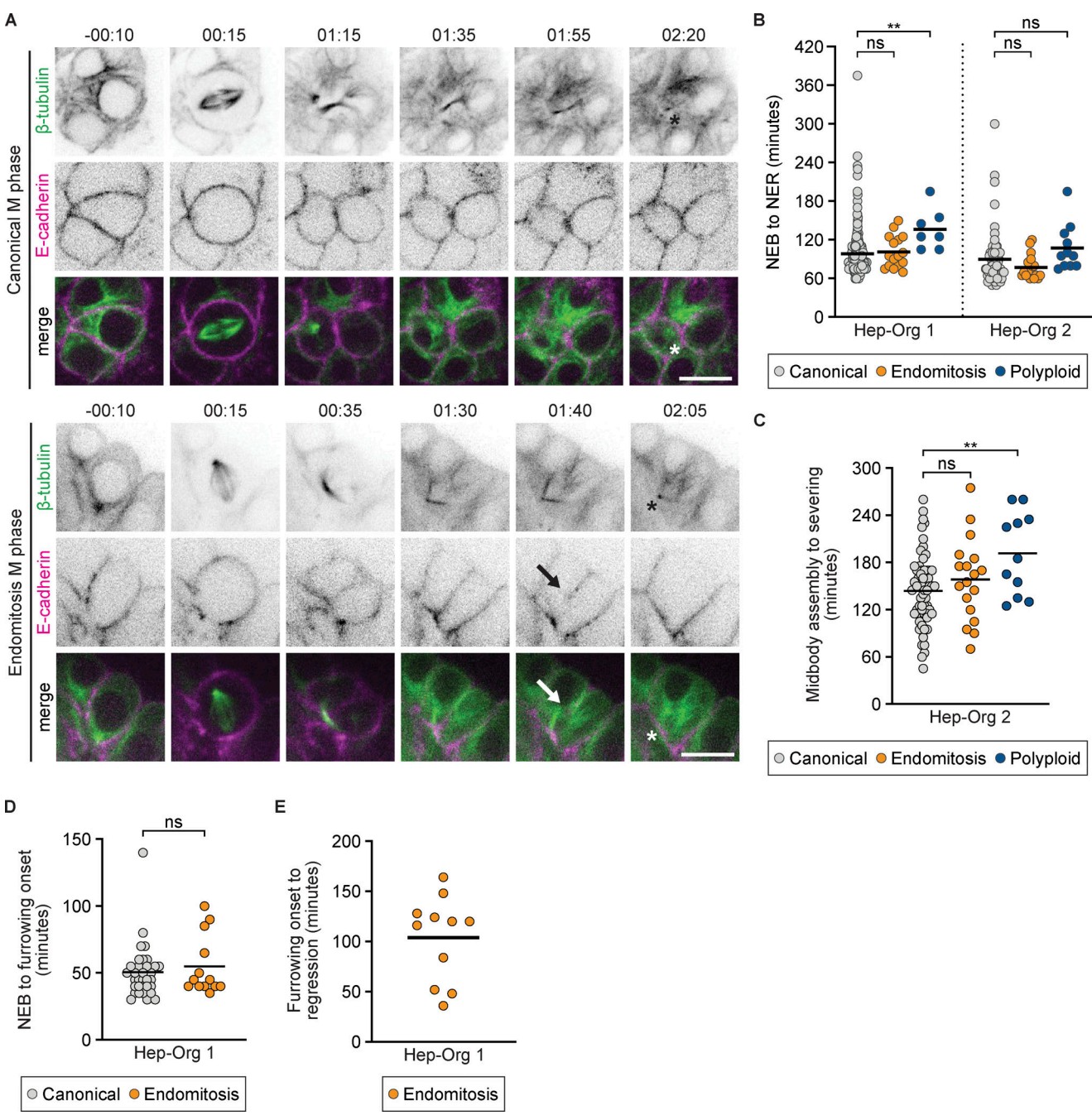

Figure 2. **Cells undergoing endomitosis have normal mitotic timings but regress their cytokinetic furrow during late M phase. (A)** Representative stills from live imaging of Tubulin-mNeon/E-cadherin-tdTomato Hep-Org 2 line showing canonical (top) and endomitosis (bottom) M phases. Stills show formation of central spindle in both canonical mitosis and endomitosis, with subsequent membrane regression in endomitosis (marked with arrow) and midbody severing (marked with asterisk). Time is relative to NEB in h:min. Scale bars represent 50 μm. Panels 3–6 showing β-tubulin are maximum projections of two z-slices. **(B)** Duration of NEB to NER in minutes for canonical, endomitosis, and polyploid M phases in Hep-Org 1 line expressing GFP-NLS/E-cadherin-tdTomato ($N$ = 9 experiments) and Hep-Org 2 line expressing Tubulin-mNeon ($N$ = 6 experiments). Individual measurements are shown for canonical (Hep-Org 1, $n$ = 204 events; Hep-Org 2, $n$ = 62 events), endomitosis (Hep-Org 1, $n$ = 16 events; Hep-Org 2, $n$ = 18 events), and polyploid (Hep-Org 1, $n$ = 7 events; Hep-Org 2, $n$ = 11 events) M phases. Black bars indicate mean (ns = not significant, **$P < 0.01$, Student's $t$ test, two-tailed). **(C)** Duration from midbody assembly to severing in minutes for canonical ($n$ = 62 events), endomitosis ($n$ = 18 events), and polyploid ($n$ = 11 events) M phases in Hep-Org 2 line expressing Tubulin-mNeon ($N$ = 5 experiments). Individual measurements are shown with mean (black bar) (**$P < 0.01$, Student's $t$ test, two-tailed). **(D)** Duration from NEB to furrowing onset for canonical ($n$ = 32 events) and endomitosis ($n$ = 13 events) M phases in Hep-Org 1 line expressing GFP-NLS/E-cadherin-tdTomato ($N$ = 5 experiments). Individual measurements are shown with mean (black bar) (ns = not significant, Student's $t$ test, two-tailed). **(E)** Duration from furrowing onset to cytokinetic regression in endomitosis M phase in Hep-Org 1 line expressing GFP-NLS/E-cadherin-tdTomato ($n$ = 11 events). Individual measurements are shown with mean (black bar).

2018; Hesse et al., 2018). To determine how human hepatocytes inhibit cell division during endomitosis, we performed live imaging on Hep-Org lines with endogenously tagged β-tubulin-mNeonGreen and E-cadherin-tdTomato (from here on Tubulin-mNeon/E-cadherin-tdTomato) to visualize the formation of the mitotic spindle and characterize cytokinetic events. Similar to canonical M phases, we observed central spindle assembly, furrow ingression, and midbody formation in all cases of endomitosis (Fig. 2 A, n = 18, Videos 1 and 2). However, during late stages of endomitosis M phase, the cell membrane disconnects from the midbody and regresses. Despite being disconnected from the cell membrane, midbody structures remain stable after cytokinetic regression and undergo normal severing during M phase exit (Fig. 2 A).

To determine whether there are any differences in mitotic progression between cells undergoing endomitosis or canonical M phases, we compared mitotic timings using the Tubulin-mNeon Hep-Org 2 line. We found that the duration from NEB to nuclear envelope reformation (NER) is very similar between canonical and endomitosis M phases (Fig. 2 B). Furthermore, despite membrane regression, there is no difference in timing between midbody formation and disassembly in endomitosis compared with canonical M phases (Fig. 2 C). In 60% of canonical and 83% of endomitosis M phases, we observe bent midbodies during telophase (n = 12/20 for canonical M phases, and n = 15/18 for endomitosis M phases; see example of a bent midbody in the endomitosis example in Fig. 2 A). For polyploid M phases, NEB–NER duration is 1.5-fold higher than canonical and endomitosis M phases. This is not unexpected, considering these cells contain twice as many chromosomes and centrosomes, which will likely delay their mitotic progression.

Together, our data suggest that canonical and endomitosis M phases in human hepatocytes are very similar during early stages of mitosis, but that endomitotic cells inhibit their division at a late step of cytokinesis. To rule out the possibility that cells undergoing endomitosis exhibited subtle differences in early cytokinetic events, we also measured the timing of cytokinesis onset and the time between initiation and completion of cytokinetic ingression using the GFP-NLS/E-cadherin-tdTomato Hep-Org line. We found no significant differences in the timing from NEB to cytokinesis onset in cells undergoing canonical versus endomitosis M phase (Fig. 2 D). Moreover, when imaging Hep-Orgs with 5-min intervals, all cells completed cytokinetic ingression within two time frames after cytokinetic onset (n = 34 for canonical and n = 13 for endomitosis M phase). The timing of cytokinetic regression was more variable between cells undergoing endomitosis, ranging between half an hour and 2.5 h (Fig. 2 E). Taken together, we find that endomitosis in human hepatocytes is associated with alterations in late, but not early, cytokinetic events.

### Cells undergoing endomitosis lose membrane anchoring to the midbody

To understand the underlying cause of membrane regression during endomitosis, we further examined cytokinetic events, zooming in on cytokinetic furrows and midbody structures using immunofluorescent (IF) staining in the E-cadherin-

tdTomato Hep-Org 1 line. We first determined whether we could identify endomitosis M phases in fixed Hep-Orgs. When looking at static images, ingression of the cleavage furrow during early telophase looks similar to a regressed membrane in endomitosis, making it difficult to distinguish the two by solely looking at the membrane. Nonetheless, by staining for α-tubulin and co-staining with DAPI, we could identify cells with regressed membranes in late telophase by the presence of a midbody structure and decondensed DNA. In contrast, cells in early telophase have condensed DNA and a broader central spindle structure. When looking at all cells in late telophase, we found that 13 out of 169 cells (7.69%) had regressed membranes, which is in line with the percentage of endomitosis M phases that we found using live imaging (Fig. 1 E). Thus, we can use IF staining on fixed Hep-Orgs to further analyze regressed structures during endomitosis M phase.

We first investigated whether endomitotic regression could be a consequence of the activation of an abscission checkpoint. Specifically, presence of chromosome bridges in the cleavage plane activates the abscission checkpoint and can result in cytokinetic regression if the chromosome bridges are not resolved (Steigemann et al., 2009). Therefore, we determined whether we could detect any DNA in the cleavage plane of cells undergoing late telophase in E-cadherin-tdTomato Hep-Orgs stained with DAPI and anti-α-tubulin. In 179/179 cases, including 13 cells undergoing endomitotic regression, we could not detect any DNA in the cleavage plane. To exclude that cells in late anaphase contained ultrafine DNA bridges that would not be visible by DAPI staining, we also performed IF staining for RIF1, a factor that localizes to ultrafine DNA bridges in anaphase and is required for their resolution (Hengeveld et al., 2015). Consistent with previous analyses of non-transformed, non-stressed cells, we found that early anaphase cells often contained dim RIF1-positive thread-like structures, which likely reflect naturally occurring ultrafine DNA bridges caused by catenated centromeric DNA (Fig. S1). However, in late anaphase and telophase, all cells were negative for RIF1 staining in between the segregated DNA, showing that all ultrafine DNA bridges had been resolved (n = 57/57, Fig. S1). Together, this makes it unlikely that membrane regression in endomitosis is due to the presence of chromosome bridges.

We next focused on cells that were in the process of membrane regression to understand how cytokinesis is inhibited in endomitosis. During canonical M phase, actomyosin ring contraction generates a cleavage furrow that partitions the mother cell into two. Once ingression is complete, several protein complexes anchor the cell membrane to the microtubules of the midbody to stabilize the cytokinetic furrow (Mierzwa and Gerlich, 2014). Subsequently, actin filaments of the actomyosin ring disassemble and the cell undergoes abscission, in which the endosomal sorting complex required for transport III (ESCRT-III) splits the plasma membrane, giving rise to two daughter cells. In addition to its essential role in membrane abscission, the ESCRT-III machinery is also required to initiate microtubule severing by targeting the microtubule-severing enzyme Spastin to the midbody (Mierzwa and Gerlich, 2014; Yang et al., 2008). Since we did not detect any abnormalities in midbody assembly,

furrow ingression, or microtubule severing (see Fig. 2 A), we wondered whether membrane anchoring is impaired during endomitosis M phases. In line with a defective anchoring of the membrane to the midbody, we found that the membrane was detached from either both sides ($n$ = 6/9) or from one side ($n$ = 3/9) of the midbody in late-stage endomitotic regressions.

Previous work has shown that defects in membrane anchoring of the midbody results in a late-cytokinetic regression (Lekomtsev et al., 2012; Kechad et al., 2012), similar to what we observe during endomitosis. To investigate whether there are any differences in the localizations of membrane anchoring proteins during endomitosis M phase, we performed IF staining of four key membrane anchoring components: RacGAP1, Anillin, Septin 9 (SEPT9), and citron Rho-interacting kinase (CIT-K). We focused our analysis on cells in early telophase and cells in late telophase with either ingressed cleavage furrows or regressed membranes, the latter reflecting cells in endomitosis. We categorized the localizations of RacGAP1, Anillin, SEPT9, and CIT-K during late anaphase and late telophase to determine if there were any abnormalities in regressed structures (Figs. 3 and 4). The centralspindlin component RacGAP1 was present at the central spindle in late anaphase and localized to the midbody ring in late telophase. RacGAP1 was not present at the membrane during late anaphase, but its localization overlapped with the E-cadherin signal in late telophase cells with ingressed cleavage furrows (Fig. 3 A and Fig. 4 A). In regressed structures, RacGAP1 retained its normal localization to the midbody ring but was no longer associated with the regressed cell membrane (Fig. 3 A and Fig. 4 A). Similarly, Anillin, which initially localized to the cleavage furrow in late anaphase, maintained midbody localization in regressed structures but could no longer be detected at the cell membrane (Fig. 3 B and Fig. 4 B). SEPT9 localized to the membrane in 83% of ingressed late telophase structures and was also found on the midbody ring and arms (Fig. 3 C and Fig. 4 C). In regressed structures, SEPT9 was still present on the midbody ring and arms, but was mostly absent from regressed membranes ($n$ = 5/7 cells, Fig. 3 C and Fig. 4 C). Finally, analysis of CIT-K localization revealed two distinct localization patterns on the midbody: either restricted to the midbody ring, as has been described before (Fig. 3 D and Fig. 4 D) (Gai et al., 2011; Hu et al., 2012; Watanabe et al., 2013), or along the whole midbody (Fig. S2). The broad midbody localization was observed in 37% of ingressed late telophase structures and 100% of regressed structures (Fig. 4 D). Furthermore, CIT-K colocalized with E-cadherin in all early telophase and ingressed structures, but was not found on the cortex in cells undergoing endomitotic regression (Fig. 3 D and Fig. 4 D).

Taken together, we find that the membrane anchoring proteins RacGAP1, Anillin, SEPT9, and CIT-K are present on the midbodies of cells undergoing endomitosis M phase, but lose their association with the membrane. Furthermore, we find that CIT-K localizes to a broader region on the midbody than has previously been described in cells undergoing cytokinesis, which could indicate an altered or impaired function of CIT-K in endomitosis.

## WNT signaling inhibits binucleation of human hepatocytes in an E2F7/8-dependent manner

Next, we wondered what determines the decision to undergo canonical or endomitosis M phase in hepatocytes. In the liver, hepatocytes are organized in hexagonal lobules. Along the rows of hepatocytes, from the central vein in the middle to the periportal veins at the corners, hepatocytes experience gradients in nutrients, oxygen, and cytokines, and exhibit distinct patterns of gene expression (Jungermann and Katz, 1982; Jungermann et al., 1982; Benhamouche et al., 2006; Burke et al., 2009; Wang et al., 2015; Halpern et al., 2017). One of these gradients is the WNT gradient, which is secreted from the central vein and has been recently shown to promote canonical mitosis in mouse hepatocytes through Tbx3-mediated repression of E2F7 and E2F8 (Jin et al., 2022).

Under normal culturing conditions, Hep-Orgs were kept in an active WNT state by the presence of CHIR99021, a GSK-3 inhibitor and potent activator of WNT signaling. To test whether WNT signaling regulates binucleation in human hepatocytes, we removed CHIR99021 from the culture medium for 3 days and quantified the percentage of binucleated cells in Hep-Orgs. Strikingly, we found a twofold increase in the percentage of binucleated cells after 3 days of CHIR99021 removal from the medium (Fig. 5 A). Additionally, Hep-Orgs cultured without CHIR99021 proliferated slower and changed morphology, growing in a round shape with a large lumen (Fig. 5 B). This suggests that, similar to what has been described in murine hepatocytes, WNT signaling supports proliferation and influences hepatocyte differentiation and/or function.

To determine if WNT-mediated repression of binucleation in Hep-Orgs is dependent on E2F7 and E2F8, as has been reported in mouse hepatocytes (Celton-Morizur et al., 2009; Margall-Ducos et al., 2007; Pandit et al., 2012), we used the CRISPR-based cytosine base editor system (Komor et al., 2016; Koblan et al., 2018) to introduce premature stop codons in the endogenous *E2F7* and *E2F8* loci (Fig. 5 C). The mutant lines showed differences in proliferation and morphology compared with wildtype Hep-Orgs: the E2F7$^{Q206X}$ line proliferated slower and exhibited a dense morphology, while the E2F8$^{Q462X}$ line grew at a comparable rate with a similar bunch-of-grapes morphology as the wildtype line (Fig. 5 D). We confirmed by quantitative reverse transcription PCR (RT-qPCR) that the introduction of a premature stop codon resulted in nonsense-mediated decay of the respective genes (Fig. 5 E) and also found that the expression of *E2F8* is increased in the E2F7$^{Q206X}$ line, while expression of *E2F7* is decreased in the E2F8$^{Q462X}$ line. In normal culture conditions, the percentage of binucleated cells was lower in the E2F8$^{Q462X}$ line than in wildtype but was comparable between E2F7$^{Q206X}$ and wildtype Hep-Org lines (Fig. 5 F). Together, these suggest that knockdown of E2F7 or E2F8 individually is not sufficient to completely prevent binucleation, but that E2F7 and E2F8 function redundantly to promote binucleation.

Strikingly, we did not see an increase in the percentage of binucleated cells upon removal of CHIR99021 in the E2F7$^{Q206X}$ and E2F8$^{Q462X}$ mutant Hep-Org lines, which could indicate that E2F7 and E2F8 act downstream of WNT signaling to induce binucleation (Fig. 5 F). To exclude that the differences in the

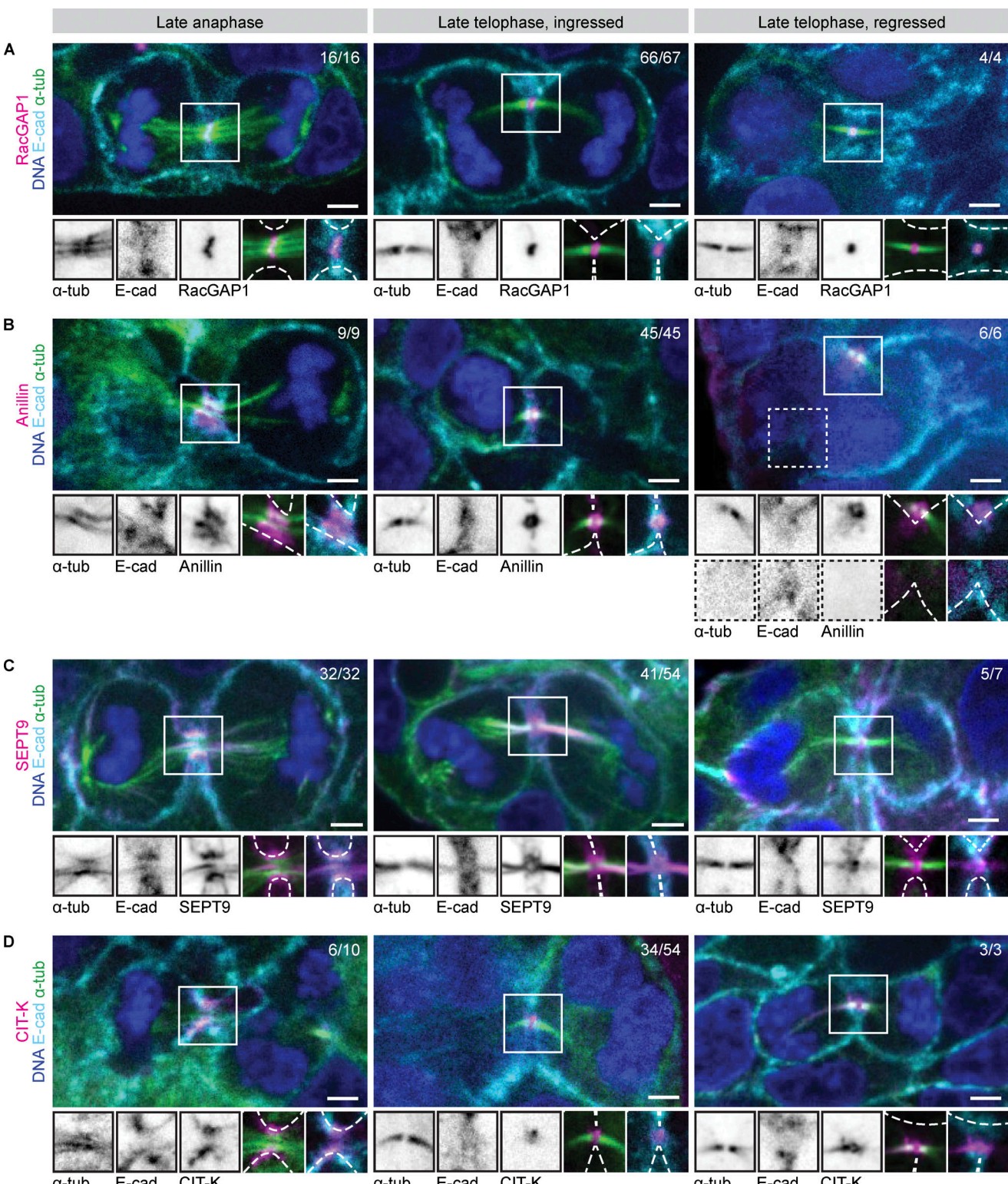

Figure 3. **Membrane association of membrane anchoring proteins is lost during cleavage regression in endomitosis. (A–D)** Representative stills of IF experiments showing DAPI, α-tubulin, E-cadherin, and membrane-anchoring protein (A) RacGAP1, (B) Anillin, (C) SEPT9, or (D) CIT-K in late anaphase and late telophase, with either ingressed or regressed cleavage furrows. Hep-Org 1 line expressing E-cadherin-tdTomato was used for IF stainings. Scale bars represent 3 μm. Close-ups show single- or double-channel images of marked regions of interest (white box). Dashed lines represent membrane outline. All images show one plane in the middle of the midbody. Numbers in the top right show quantification of displayed localization. More detailed quantifications of RacGAP1, Anillin, SEPT9, and CIT-K localizations during the different stages are shown in Fig. 4.

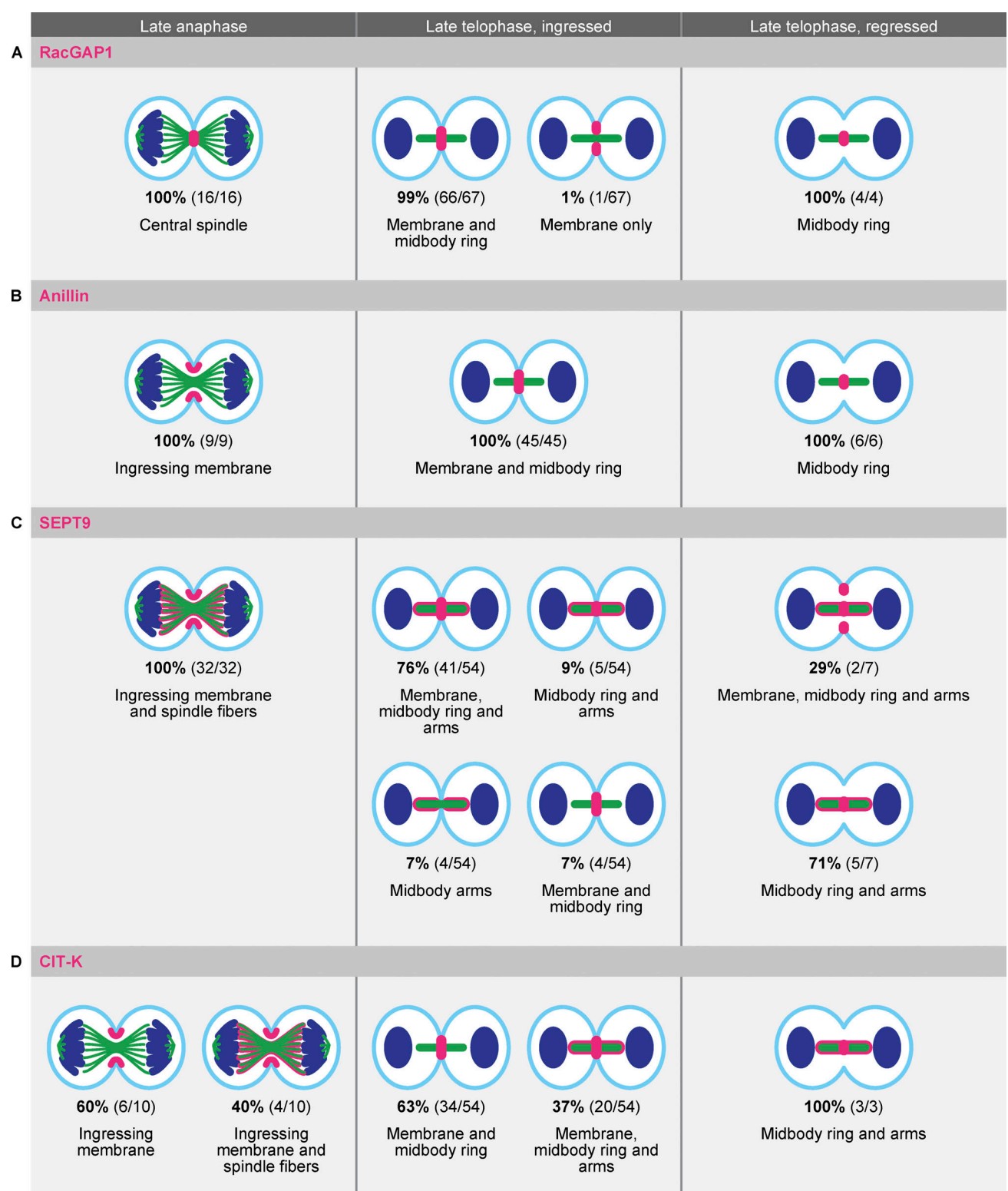

Figure 4. **Localization of membrane anchoring proteins in Hep-Orgs cells undergoing M phase. (A–D)** Schematic overview of the different localizations of (A) RacGAP1, (B) Anillin, (C) SEPT9, and (D) CIT-K (magenta) in late anaphase and late telophase with either ingressed or regressed cleavage furrows. Cell membranes are depicted in cyan, DNA in blue, and microtubules in green. The percentages and numbers show how often the depicted localization of the membrane anchoring protein was observed.

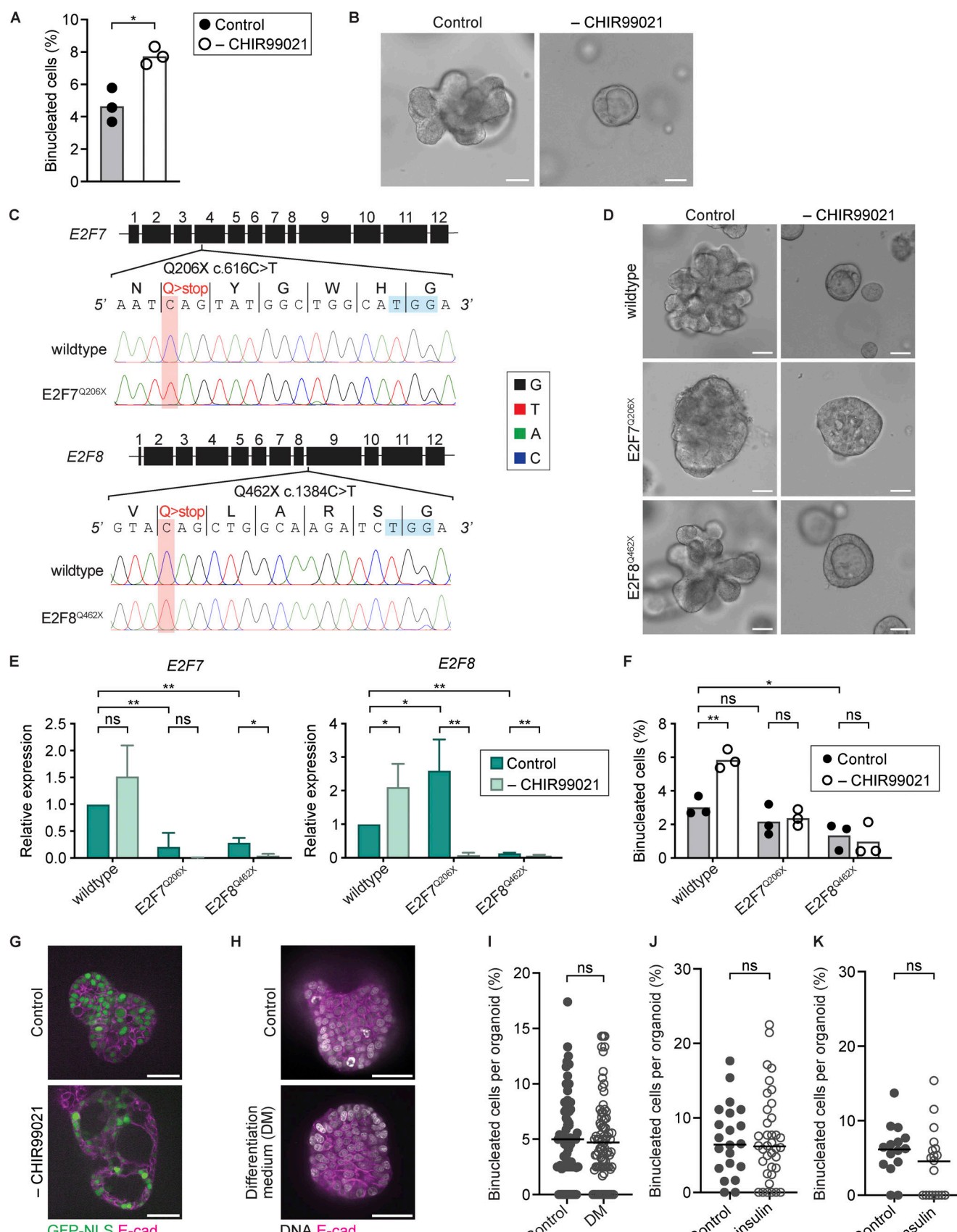

Figure 5. **WNT signaling inhibits binucleation of human hepatocytes in an E2F7/8-dependent manner. (A)** Percentage of binucleated cells in Hep-Org 1 line with GFP-NLS and stained with CellMask Orange in control medium and after 3 days of CHIR99021 removal (*N* = 3 experiments, 200–300 cells analyzed

per experiment, *P < 0.05, Student's *t* test, two-tailed). **(B)** Representative brightfield images of Hep-Org 1 in control medium and after 3 days of CHIR99021 removal. Scale bars represent 50 μm. **(C)** Base-editing strategy for introduction of a premature stop codon in *E2F7* and *E2F8* open reading frames with Sanger sequencing chromatographs of wildtype and mutant alleles, confirming homozygous base changes (highlighted in red) in E2F7$^{Q206X}$ and E2F8$^{Q462X}$ Hep-Org 1 lines. Protospacer adjacent motif (PAM) sequences are highlighted in blue. **(D)** Representative brightfield images of wildtype, E2F7$^{Q206X}$, and E2F8$^{Q462X}$ Hep-Org 1 lines in control medium and after 3 days of CHIR99021 removal. Scale bars represent 50 μm. **(E)** Relative expression of *E2F7* or *E2F8* in wildtype and mutant lines as measured by RT-qPCR in control medium and after 3 days of CHIR99021 removal (*N* = 3 experiments, ns = not significant, *P < 0.05, **P < 0.01, Student's *t* test, two-tailed). Error bars represent standard deviation. **(F)** Percentage of binucleated cells in wildtype and mutant lines expressing GFP-NLS and stained with CellMask Orange in control medium and after 3 days of CHIR99021 removal. Each dot represents the average percentage of binucleated cells per experiment (*N* = 3 experiments, 200–300 cells analyzed per experiment, ns = not significant, *P < 0.05, **P < 0.01, Student's *t* test, two-tailed). **(G)** Representative images of Hep-Org 1 line expressing GFP-NLS (green) and E-cadherin-tdTomato (magenta) grown in control medium and after 3 days of CHIR99021 removal. Scale bars represent 50 μm. **(H)** Representative images of Hep-Org 1 line expressing E-cadherin-tdTomato (magenta) and stained with DRAQ5 (gray) to visualize DNA, after growth for 7 days in either control medium or differentiation medium (DM). Scale bars represent 50 μm. **(I)** Percentage of binucleated cells per organoid in Hep-Org 1 line expressing E-cadherin-tdTomato and stained with DRAQ5 after growth for 7 days in either control medium (*n* = 84 organoids) or differentiation medium (DM, *n* = 83 organoids). Each dot represents the percentage of binucleated cells per organoid, and the black line represents the median (*N* = 2 experiments, 73–93 organoids analyzed per experiment, ns = not significant, Mann–Whitney test). **(J)** Percentage of binucleated cells per organoid in Hep-Org 1 line expressing GFP-NLS/E-cadherin-tdTomato grown for 1 day in control medium (*n* = 21 organoids) or in medium supplemented with insulin (*n* = 37 organoids). The black line represents the median. See material and methods section for more information on media composition (*N* = 2 experiments, ns = not significant, Mann–Whitney test). **(K)** Percentage of binucleated cells per organoid in Hep-Org 1 line expressing GFP-NLS/E-cadherin-tdTomato grown for 7 days in control medium (*n* = 15 organoids) or in medium without insulin (*n* = 19 organoids). The black line represents the median. See Materials and methods section for more information on media composition (*N* = 2 experiments, ns = not significant, Mann–Whitney test).

proportion of binucleated cells upon knockdown of E2F7 or E2F8 arise by differences in the cell-cycle distributions of cells, for example, an increase in mononucleated 4N cells, we quantified ploidy distributions upon knockdown of E2F7 or E2F8. We measured DAPI intensities of mononucleated cells in wildtype, E2F7$^{Q206X}$, and E2F8$^{Q462X}$ Hep-Org lines grown under normal conditions or upon CHIR99021 removal. Our analyses revealed no differences in ploidy distributions, suggesting that E2F7 and E2F8 specifically influence the percentage of binucleated cells upon CHIR99021 removal, rather than affecting a general cell-cycle distribution (Fig. S3). In line with previous research in rodents (Wang et al., 2015; Jin et al., 2022), these findings suggest that E2F7 and E2F8 promote binucleation downstream of WNT signaling in human Hep-Orgs.

To further examine how Hep-Org morphology and binucleation are influenced by different growth media, we grew GFP-NLS/E-cadherin-tdTomato Hep-Orgs in either a regular expansion medium or a differentiation medium, which was previously shown to result in the maturation of human Hep-Orgs (Hu et al., 2018). In contrast to cells grown in the absence of CHIR99021 (Fig. 5 G), we see no gross morphological differences in Hep-Orgs grown in differentiation medium (Fig. 5 H). Also, the percentage of binucleated cells is similar in Hep-Orgs grown in expansion or differentiation medium (Fig. 5 I). Finally, we tested whether the addition or removal of insulin, which is thought to control endomitosis in rodents (Celton-Morizur et al., 2009; Donne et al., 2020), influences Hep-Org binucleation; however, we found no differences in the percentage of binucleated cells per organoid (Fig. 5, J and K). Taken together, our results suggest that both WNT signaling and E2F7/E2F8 influence hepatocyte binucleation; however, increases in binucleation that have been observed in aging rodent livers cannot be recapitulated in vitro in human Hep-Orgs.

## Discussion

Although non-canonical cell cycles are crucial for the function of many organs, little is known about how they are initiated and executed in human cells, largely due to the unavailability of suitable systems. In the present study, we used non-transformed, healthy human Hep-Orgs to study how cells undergo non-canonical cell cycles and become binucleated. Our findings indicate that binucleated cells in Hep-Orgs arise by detachment of the cell membrane to the midbody in late cytokinesis. This late cytokinesis exit contrasts with endomitosis in rodent livers, where cells lack a central spindle and do not undergo cleavage furrow ingression (Guidotti et al., 2003; Margall-Ducos et al., 2007; Celton-Morizur et al., 2009). Despite the differences in the mechanism by which cells inhibit cell division, we uncover a conserved function of WNT signaling along with E2F7 and E2F8 in the regulation of endomitosis, suggesting that similar mechanisms control the choice between canonical versus non-canonical cell cycles in rodents and human hepatocytes.

We find that on average, 3–15% of cells in the Hep-Org lines used in this study are binucleated. The differences in percentages of binucleated cells between the two Hep-Org lines likely reflect normal biological variation as they are consistent with reported percentages of binucleated cells in fetal and neonatal human livers, ranging between 1.5 and 8% (Gahan and Middleton, 1984; Kudryavtsev et al., 1993). Notably, despite that the Hep-Org 2 line shows higher percentages of binucleated cells, this is not reflected in the percentage of cells undergoing endomitosis M phase, which is very similar between Hep-Org 1 and Hep-Org 2 lines. This may indicate that the higher percentage of binucleated cells in Hep-Org 2 is due to longer retention of binucleated cells instead of a higher rate of endomitosis. In both rodents and humans, the number of binucleated hepatocytes increases with age, with rodent livers containing on average around 80% binucleated cells (Duncan et al., 2010) and adult human livers consisting of up to 20% binucleated cells (Gahan and Middleton, 1984; Kudryavtsev et al., 1993; Toyoda et al., 2005). In mouse livers, the percentage of hepatocytes undergoing endomitosis increases drastically upon weaning; however, it is unknown what triggers endomitosis cycles in human livers (Celton-Morizur et al., 2009; Margall-Ducos et al., 2007; Pandit et al., 2012). In rodents,

changes in insulin signaling are thought to control the switch from canonical to endomitosis cycles in the liver (Celton-Morizur et al., 2009; Donne et al., 2020). It is possible that changes in hepatic metabolism also influence the choice between canonical and non-canonical M phases in human hepatocytes; however, our attempts to grow Hep-Org lines with different concentrations of insulin or in a growth medium that diminishes fetal-like properties of hepatocytes (Hu et al., 2018) did not result in an increase in the percentage of binucleated cells. It is possible that our growth media lack critical factors that promote adult-like maturation of hepatocytes. Indeed, Hep-Orgs can only be generated from fetal human livers and not adult human livers, suggesting that mature hepatocytes have distinct requirements for their survival and growth. Future research will be needed to identify the optimal growth conditions that promote hepatocyte maturation and how this influences hepatocyte binucleation.

From our live-imaging analyses of Hep-Orgs, we find that human hepatocytes undergoing endomitosis inhibit cell division during a late step in cytokinesis. Endomitotic hepatocytes display normal cytokinetic furrow ingression, but the ingressed membrane detaches from the midbody during late cytokinesis. We find that RacGAP1, Anillin, SEPT9, and CIT-K, which are involved in anchoring the midbody to the cell cortex, are present but lose their association with the cell membrane during endomitotic furrow regression. Mutation of the membrane binding domains of RacGAP1 or Anillin in insect or human cells gives rise to a late cytokinetic regression, similar to what we observe during endomitosis M phase (Lekomtsev et al., 2012; Kechad et al., 2012) Thus, it is possible that a failure to stably anchor the membrane to the midbody is driving cytokinetic regression during endomitosis. Alternatively, the loss of anchoring proteins at the regressing membrane could be a secondary consequence of other differences in the regulation of cytokinesis in endomitosis. For example, it is possible that there are differences in the posttranslational modification of cytokinesis regulators, leading to cytokinetic regression in endomitosis. Interestingly, we find that the kinase CIT-K shows a distinct localization along the midbody arms during endomitosis. In canonically dividing Drosophila and human cells, CIT-K is restricted to the midbody ring during late telophase, where it likely plays both a structural and signaling role in the connection between the cell cortex and the midbody (D'Avino, 2017). Strikingly, a broader CIT-K distribution during telophase has also been observed in human cells upon activation of Ephrin signaling, which correlates with the appearance of multinucleated, polyploid cells (Jungas et al., 2016). Furthermore, inhibition of CIT-K by Src kinase in the mouse brain promotes the formation of polyploid neurons during neocortex development, and Src kinase activity is also thought to promote mouse hepatocyte binucleation (De Santis Puzzonia et al., 2016; Jungas et al., 2016). Thus, it will be interesting to further investigate how specific kinase activities influence the rates of endomitosis in Hep-Orgs.

Our findings suggest that there are also differences in endomitosis between human and rodent hepatocytes (Guidotti et al., 2003; Margall-Ducos et al., 2007; Celton-Morizur et al., 2009), raising the question of whether cells evolved independent

mechanisms to become polyploid. This is not unlikely, as it is known that within the same cell type, endomitosis does not always occur in the same manner across different species. For instance, the majority of polyploid mouse cardiomyocytes are binucleated, having aborted cytokinesis at a late stage of M phase (Derks and Bergmann, 2020), whereas the majority of polyploid human cardiomyocytes are mononucleated and thus likely exit M phase before anaphase (Brodsky et al., 1991; Olivetti et al., 1996; Bergmann et al., 2009; Mollova et al., 2013; Windmueller et al., 2020). In rodent cardiomyocyte endomitosis, cytokinetic ingression is slow or absent, and this has been associated with aberrant localization of the cytokinesis regulators RhoA, IQ-GAP3, and Anillin (Engel et al., 2006; Leone et al., 2018). In contrast, in human cardiomyocytes undergoing endomitosis, most cytokinesis regulators seem to be normally expressed; however, the RhoA guanine exchange factor ECT2 is downregulated (Liu et al., 2019). Taken together, these findings suggest that the same cell type within different species can have distinct mechanisms of endomitosis. It will be interesting to understand whether these differences arise because these cell types, such as cardiomyocytes or hepatocytes in mammals, have evolved the ability to undergo endomitosis independently or rather because they have diverged in the mechanism of endomitosis.

Our analyses of human hepatocytes undergoing endomitosis revealed no apparent differences in early aspects of cytokinesis, such as the initiation and speed of furrowing, suggesting that human hepatocytes undergoing endomitosis have specifically altered late cytokinetic events. In rodents, hepatocytes undergoing endomitosis M phase seem to completely lack a central spindle and do not undergo any cytokinetic ingression (Guidotti et al., 2003; Margall-Ducos et al., 2007), suggesting that different cytokinesis regulators are altered in rodent hepatocyte endomitosis compared with humans. Nonetheless, our analyses suggest that there are also similarities in the regulation of endomitosis: first, we find that low levels of WNT signaling correlate with higher percentages of binucleated hepatocytes, as was previously shown in mice (Jin et al., 2022); second, we find that the atypical E2F family transcription factors E2F7 and E2F8 seem to function downstream of WNT signaling to promote binucleation, suggesting that there may be a conserved pathway to control endomitosis in mammalian livers. Nonetheless, it remains possible that WNT signaling and E2F7/8 control the number of binucleated cells independently of endomitosis, for example, by influencing the survival rate of binucleated cells. Furthermore, we cannot exclude the possibility that knockout of E2F7 or E2F8 influences the ability of cells to respond to CHIR99021 removal.

Intriguingly, we find that decreased WNT activity or knockdown of E2F7 or E2F8 gives rise to differences in morphology of Hep-Orgs, which could be indicative of cellular differentiation. Culturing of Hep-Orgs in a growth media that was previously shown to promote the expression of mature-like hepatocyte genes did not affect the percentages of binucleated cells. Similarly, addition or removal of insulin from the growth media also did not alter the percentage of binucleated cells. It is possible that other signals, which we are missing in the Hep-Org

cultures, are required to mimic the increase in endomitosis that has been observed in aging livers. Future work should be aimed at characterizing the gene expression changes that occur upon aging in human livers, as well as the effects of CHIR99021 removal and knockdown of E2F7 and E2F8 in Hep-Orgs, which will help elucidate how these different factors influence binucleation and hepatocyte differentiation.

Interestingly, E2F7 and E2F8 have not only been implicated in endomitosis but also in endoreplication cycles in trophoblast giant cells (Ouseph et al., 2012; Matondo et al., 2018; Chen et al., 2012). As initiation of endoreplication requires inhibition of mitotic entry, it is possible that E2F7 and E2F8 act as general repressors for genes involved in M phase, including genes responsible for cytokinesis. Indeed, knockout of E2F7 and E2F8 in mouse livers led to an increase in expression of M phase genes, and E2F8 has also been shown to bind to promoters of M phase genes in human cells (Pandit et al., 2012). This raises the question of how the activity of E2F7 and E2F8 is specifically required to induce binucleation and does not lead to complete M phase absence and thus endoreplicative cycles in hepatocytes. One possibility is that in endomitotic cells, activation of E2F7 and E2F8 leads to transcriptional repression of all M phase genes, but the available proteins remain stable at sufficient levels to allow mitotic entry but are not sufficient to complete cytokinesis. To test this hypothesis, one would require a detailed comparison of expression profiles between cells undergoing canonical and endomitosis M phases. However, several technical aspects currently render this challenging: first, we currently lack the knowledge to be able to predict whether a cell will undergo a canonical or endomitosis M phase, preventing isolation of this rare cell population; and second, M phase regulators are expressed at low and fluctuating levels during the cell cycle, making it difficult to compare gene expression profiles between cells. Further dissection of the pathways that drive endomitosis may allow generation of mutant hepatocytes with higher levels of endomitosis, which will facilitate a detailed comparison between cells undergoing canonical and endomitosis M phase.

Taken together, our study provides insights into the mechanism by which human cells become binucleated through non-canonical cell cycles during tissue formation. We find that endomitosis in human hepatocytes occurs by loss of membrane anchorage to the midbody during M phase and that binucleation is controlled by WNT signaling and E2F7/8 activities. As liver polyploidization is important to balance liver gene expression and proliferation (Bahar Halpern et al., 2015; Zhang et al., 2018; Matsumoto et al., 2020), understanding the modulation of non-canonical cell cycles may provide therapeutic opportunities in the treatment and prevention of diseases such as liver cancer.

### Limitations of the study

Not all of the analyses described in this study could be performed in both Hep-Org lines because Hep-Org line 2 is more difficult to maintain in culture and does not tolerate CRISPR/Cas9-based engineering well, limiting the generation of reporter lines. Although we demonstrate that endomitosis occurs by a late cytokinetic regression in both Hep-Org lines, the question remains whether this mimics how hepatocytes undergo endomitosis in the human liver. Furthermore, we inferred a function of E2F7 and E2F8 in the regulation of endomitosis through our analyses on the percentage of binucleated cells upon knockout of E2F7 and E2F8; however, we did not examine whether the expression of cytokinesis regulators is altered in these conditions. mRNA expression analyses of wildtype, E2F7$^{Q206X}$, and E2F8$^{Q462X}$ mutant Hep-Org lines grown with or without CHIR99021 may help identify transcriptional targets that drive endomitosis in Hep-Orgs.

## Materials and methods

### Organoid culture

Hep-Orgs used in this study were previously generated from human fetal livers that were obtained from Leiden University Medical Centre (MC) under the approval of the Dutch Ethical Medical Council (Leiden University MC) (Hu et al., 2018; Artegiani et al., 2020). All organoids were grown in Cultrex reduced growth factor basement membrane extract (BME), type 2 (#3533-001; R&D Systems). Culturing medium was refreshed every 2 to 3 days, and organoids were split 1:6 or 1:8 every 7 days as previously described for Chol-Orgs (Broutier et al., 2016) and Hep-Orgs (Hu et al., 2018). Expansion culture medium components are listed in Table S1. Differentiation medium was prepared by adding 1 μM dexamethasone (#D4902; Sigma-Aldrich) and 10 ng/μl oncostatin M (#295-OM; R&D Systems) to Hep-Org expansion medium as described previously (Hu et al., 2018). In experiments testing the effect of insulin, insulin (#I9278; Sigma-Aldrich) was added for 1 day to Hep-Org expansion medium with a final concentration of 20 μg/ml, and medium without insulin was added for 7 days and was made by replacing the B-27 Supplement minus vitamin A of the expansion medium by B-27 Supplement minus insulin (#A1895601; Gibco). Since B-27 Supplement minus insulin contains vitamin A, B-27 Supplement (#17504044; Gibco) was used as a control in these experiments.

### Generation of organoid lines

For lines expressing GFP-NLS, organoids were transduced with lentivirus containing pHR-sfGFP-NLS using the sandwich method adapted from Maru et al. (2016). In short, organoids were dissociated using TrypLE (#12605; Gibco) to small clumps and plated on a layer of BME type 2 (cat. no. 3533-001; R&D Systems) and incubated overnight in culture medium supplemented with 10 μg/ml polybrene (#sc-134220; Santa Cruz Biotechnology) containing concentrated lentiviral supernatant (Centriprep centrifugal unit—10 kDa cutoff, #4305; Sigma-Aldrich). The next day, the culture medium was replaced and a second layer of BME was added to transduced organoids to allow recovery, after which organoids were cultured normally. The generation of lines expressing endogenous Tubulin-mNeon and/or endogenous E-cadherin-tdTomato was performed using CRISPaint as previously described (Schmid-Burgk et al., 2016; Artegiani et al., 2020). Premature stop codons in *E2F7* and *E2F8* were introduced in the Hep-Org 1 line using CRISPR base editing as previously described (Geurts et al., 2020). All guide RNA (gRNA) sequences are listed in Table S2. Clonal organoid lines were obtained by selecting organoids grown from single

cells following digestion with TrypLE. For genotyping, organoids were lysed at 65°C for 15 min followed by enzyme inactivation at 95°C for 5 min in lysis buffer containing 50 mM KCl (#0509; Avantor), 2.5 mM MgCl$_2$ (#0162; Avantor), 10 mM Tris-HCl pH 8.3 (#15504-020; Invitrogen), 0.45% IGEPAL CA-630 (#I8896; Sigma-Aldrich), 0.45% Tween-20 (#P1379; Sigma-Aldrich), and 1 mg/ml Proteinase K (#P2308; Sigma-Aldrich), followed by PCR amplification using Q5 High-Fidelity DNA polymerase (#M0491; New England Biolabs) and Sanger sequencing (Macrogen Europe). Genotyping primers are listed in Table S2.

## RNA isolation and RT-qPCR
RNA was extracted from organoids using TRIzol extraction (#15596018; Invitrogen). cDNA synthesis was performed using iScript cDNA synthesis kit (#1708891; Bio-Rad). qPCR was performed using iQ SYBR Green Supermix (#1708882; Bio-Rad) in a CFX96 Real-Time PCR Detection System (Bio-Rad). qPCR primers for *E2F7* and *E2F8* were designed using NCBI Primer-BLAST to target regions upstream of the target base for CRISPR base editing and are listed in Table S2. Quantitative qPCR data was normalized to GAPDH, and expressions were calculated relative to values of wildtype organoids grown in control medium.

## Live imaging, immunofluorescence, and microscopy
For live-cell imaging, organoids were plated in 3D onto ibidi 8-well chambered coverslips with ibiTreat (#80826; ibidi) 1 day prior to imaging. Imaging was performed using a Nikon TiE microscope equipped with Borealis illumination unit (Andor), CSU-W1 spinning disk scanner unit (Yokogawa), and iXon-888 Ultra EMCCD camera (Andor) using a UPLSAPO-S 30× silicone objective (Olympus) with 1.5× optical zoom in an environmentally controlled chamber held at 37°C with 5% CO$_2$. Organoids were imaged using NIS-elements software (Nikon) for 24–72 h with 5-min intervals, with a total of 12 z-slices spaced 4 µm apart per position.

To quantify the number of binucleated cells, organoids expressing GFP-NLS were grown in 3D on ibidi 8-well chambered coverslips (#80826; ibidi) and incubated with CellMask Orange (1:2,000, #C10045; Invitrogen) for 20 min, after which the staining solution was removed and fresh culture medium was added. Organoids were imaged immediately using the same spinning disk microscope system as described above.

For IF staining, organoids were plated in 3D onto ibidi 8-well chambered coverslips (#80826; ibidi) 3 days prior to fixation. Organoids were fixed in the BME droplets in 4% paraformaldehyde in PBS (#RT15710; Electron Microscopy Sciences) for 10 min, followed by permeabilization and blocking in blocking buffer containing 1% bovine serum albumin (#A3294; Sigma-Aldrich), 10% DMSO (#D8418; Sigma-Aldrich), and 2% Triton-X100 (#T8787; Sigma-Aldrich) in PBS. Organoids were incubated overnight at 4°C with rat anti-α-tubulin (1:1,000, #NB600-506; Novus Biologicals), goat anti-RacGAP1 (1:1,000, #ab2270; Abcam), mouse anti-anillin (1:500, #MABT96; Sigma-Aldrich), mouse anti-CIT-K (1:500, #611376; BD Biosciences), rabbit anti-SEPT9 (1:1,000, #HPA042564; Sigma-Aldrich), or rabbit anti-RIF1 (1:1,000, #A300-568A; Bethyl Laboratories) in a blocking buffer. For RIF1 stainings, 0.25% glutaraldehyde (#G6257; Sigma-Aldrich) was added during fixation, after which organoids were washed three times for 10 min with 1% sodium borohydride (#452882; Sigma-Aldrich), followed by permeabilization and blocking with 0.5% Triton-X100 (#T8787; Sigma-Aldrich) in the blocking buffer. After incubation with primary antibodies, organoids were washed in PBS followed by overnight incubation with species-matched secondary antibodies conjugated to Alexa fluor dyes (Invitrogen), after which samples were washed thoroughly with PBS and co-stained with DAPI (1:1,000, #32670; Sigma-Aldrich). Organoids were rinsed in PBS and imaged in PBS using a Plan Apo λ 60× oil objective (#MRD01605; Nikon) on a Nikon Ti2 microscope equipped with NIS-elements software (Nikon), an L6Cc illumination unit (Oxxius), a CSU-X1 spinning disk scanner unit (Yokogawa), and a C11440-22C camera (Hamamatsu).

For DAPI quantifications, Hep-Orgs were grown with or without CHIR99021 (#4423; Tocris) in the culture medium. After 3 days, organoids were harvested, dissociated using TrypLE (#12605; Gibco), and spun on slides by Cytospin (Thermo Fisher Scientific). Slides were stained with DAPI (1:500, #32670; Sigma-Aldrich) and Phalloidin 647 (1:400, #A22287; Invitrogen) at room temperature for 30 min. Slides were mounted with Prolong Gold antifade (#P36934; Invitrogen).

For differentiation experiments, Hep-Orgs expressing E-cadherin-tdTomato were grown in 3D on ibidi 8-well chambered coverslips (#80826; ibidi) with control or differentiation medium. After 7 days, DRAQ5 (1:2,000, #424101; Biolegend) was added to the medium, after which organoids were immediately imaged.

Images of DAPI stainings and differentiation experiments were taken using a Plan Apo λ 40× objective (#MRD00405; Nikon) on a Nikon Ti2 microscope equipped with a L6Cc illumination unit (Oxxius), CSU-X1 spinning disk scanner unit (Yokogawa), C11440-22C camera (Hamamatsu), and NIS-elements software (Nikon). For DAPI stainings, 20 z-slices were taken with a spacing of 0.6 µm.

For overview images of organoid mutant lines, brightfield images were taken on an EVOS FL microscope (#AMF4300; Invitrogen).

## Data analysis
All images were processed and analyzed using FIJI (Schindelin et al., 2012). In our analyses, only organoids containing upwards of 10 cells were included. For quantification of percentages of binucleated cells, only healthy-looking cells with round nuclei were included in the analyses. For analysis of M phase types, the Mitotic Scoring Tool macro (Bodor, 2021) was used to aid M phase annotation. For quantification of DAPI stainings, integrated density of the DAPI signal was measured using the sum of nuclear z-slices. Binucleated cells were excluded from these analyses. DAPI signal was normalized to the median of each sample to account for technical differences between experiments. Graphs and statistical analysis were conducted in GraphPad Prism version 9.5.1.

## Online supplemental material

Fig. S1 shows the absence of RIF1-positive ultrafine DNA bridges during late anaphase and telophase. Fig. S2 shows IF staining with atypical CIT-K localization on the entire midbody. Fig. S3 shows quantification of the DAPI signal in wildtype and mutant organoid lines grown with or without CHIR99021. Table S1 lists organoid culture medium components. Table S2 lists oligo sequences of gRNAs, primers used for genotyping, and primers used for qPCRs. Videos 1 and 2 show canonical M phase and endomitosis M phase, respectively.

## Data availability

Raw data underlying this study is available from the corresponding author upon request.

## Acknowledgments

We thank H. Hu, B. Artegiani, D. Hendriks, M. Geurts, and S. van den Brink (Hubrecht Institute, Utrecht, Netherlands) for organoids, reagents, advice, and technical assistance. We thank S. Lens (University Medical Center Utrecht, Utrecht, Netherlands) for the RIF1 antibody, A. Chaigne (Utrecht University Utrecht, Netherlands) for advice and reagents, A. de Graaff and P. Toonen from the Hubrecht Imaging Centre for microscopy support, D. Bodor for the FIJI macro for M phase quantifications, and G. Kops (Hubrecht Institute) and M. Gloerich (University Medical Center Utrecht) for advice throughout the project and comments on the manuscript.

This work was supported by funding from the Cancer Genomics Centre (cancergenomics.nl).

Author contributions: G.S. Darmasaputra: Conceptualization, Formal analysis, Investigation, Visualization, Writing—original draft; C.C. Geerlings: Formal analysis, Investigation, Visualization, Writing—review & editing; S.M. Chuva de Sousa Lopes: Investigation, Project administration, Resources, Writing—review & editing; H. Clevers: Methodology, Resources, Writing—review & editing; and M. Galli: Conceptualization, Formal analysis, Funding acquisition, Supervision, Visualization, Writing—original draft, Writing—review & editing.

Disclosures: H. Clevers reported personal fees from F. Hoffmann-La Roche Ltd., Basel, Switzerland, outside the submitted work; in addition, H. Clevers had a patent to Application PCT/EP2019/082618 events 2019-11-26 Application filed by Koninklijke Nederlandse Akademie Van Wetenschappen 2019-11-26 Priority to US17/296,049 2020-06-04 Publication of WO2020109324A1 licensed "HUB, Utrecht, NL"; and "I am currently an employee of F. Hoffmann-La Roche Ltd. in Basel, Switzerland, where I am member of the Extended Executive Board and head Pharma Research & Early Development. The company has no involvement in the work published in this manuscript." No other disclosures were reported.

Submitted: 4 March 2024

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

# Supplemental material

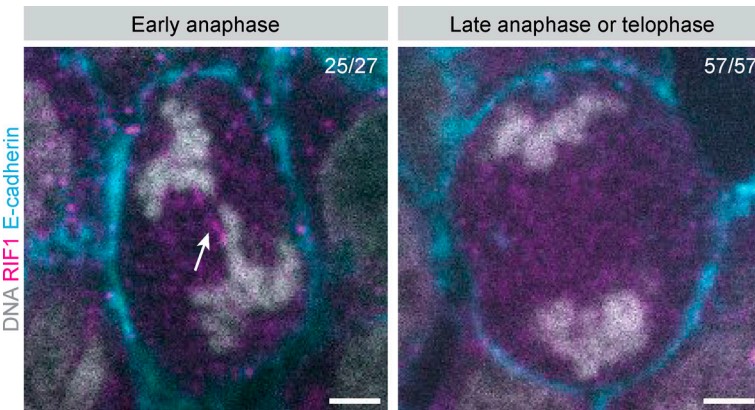

Figure S1. **Absence of RIF1-positive ultrafine DNA bridges in late anaphase and telophase.** Representative images of RIF1 staining in Hep-Org 1 expressing E-cadherin-tdTomato. RIF1 can be detected on ultrafine DNA bridges during early anaphase (left image, white arrow) but is absent in all late anaphase and telophase structures (right). Hep-Org 1 line expressing E-cadherin-tdTomato was grown in 3D, fixed, and stained for DAPI (gray) and RIF1 (magenta). Scale bars represent 3 µm. Numbers in the top right corner indicate the number of cells with RIF1-positive ultrafine DNA bridges during early anaphase (left) and the number of cells with no RIF1 staining in late anaphase or telophase (right).

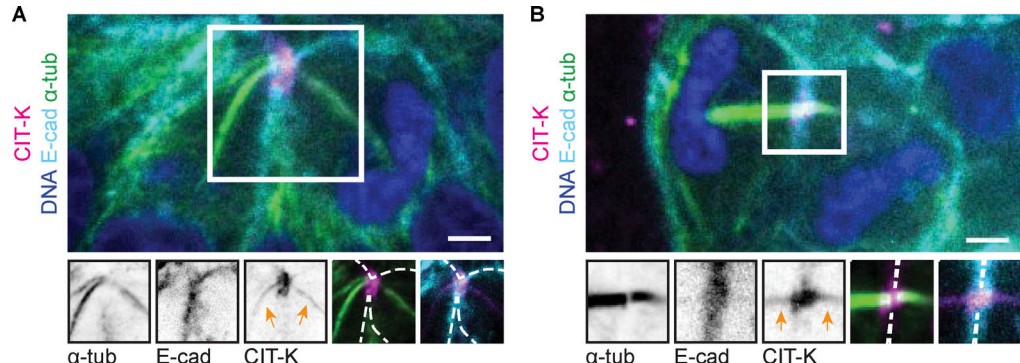

Figure S2. **Atypical CIT-K localization on microtubules during late anaphase and telophase. (A and B)** Representative images of IF experiments showing atypical CIT-K localization (magenta) with microtubules (green) in (A) early and (B) late telophase. Scale bars represent 3 µm. Close-ups show single- or double-channel images of marked regions of interest (white box). Orange arrows point to CIT-K signal in midbody arms. Dashed lines represent the membrane outline. All images show one plane in the middle of the midbody. CIT-K localization to the midbody arms was observed in 37% of ingressed late telophase structures and 100% of regressed structures.

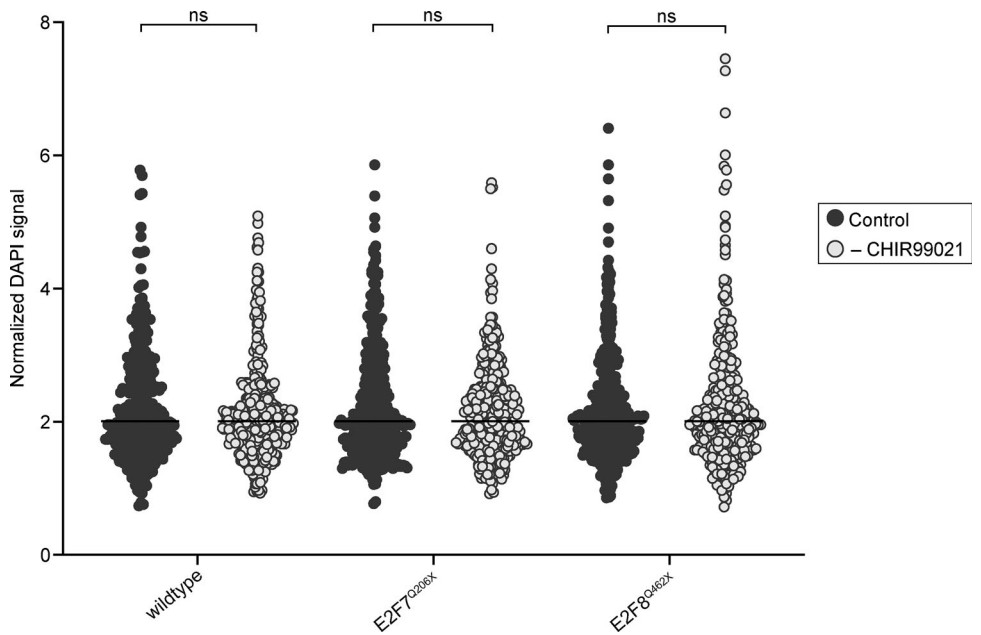

**Figure S3. Similar ploidy distributions in wildtype, E2F7$^{Q206X}$, and E2F8$^{Q462X}$ Hep-Org lines.** Quantification of DAPI signal in wildtype, E2F7$^{Q206X}$, and E2F8$^{Q462X}$ Hep-Org lines grown in culture medium or after 3 days of CHIR99021 removal. Organoids were grown in 3D, trypsinized, cytospun onto slides, fixed, and stained with DAPI. DAPI signal was normalized to the median (black line) of each sample (between 350 and 500 cells analyzed per sample, $N$ = 2 experiments, ns = not significant, Mann–Whitney test).

Video 1. **Live imaging of a canonical M phase.** Live imaging of Hep-Org 2 line expressing Tubulin-mNeon (green) and E-cadherin-tdTomato (magenta) showing a canonical M phase. Overlay of channels is shown in color (leftmost panel) next to single channels in gray (middle panel showing tubulin, right panel showing E-cadherin). Images were taken every 5 min. Time (in minutes) from NEB is indicated in the top left corner. Video is related to Fig. 2 A. Playback speed: 7 frames per second.

Video 2. **Live imaging of an endomitosis M phase.** Live imaging of Hep-Org 2 line expressing Tubulin-mNeon (green) and E-cadherin-tdTomato (magenta) showing an endomitosis M phase. Overlay of channels is shown in color (leftmost panel) next to single channels in gray (middle panel showing tubulin, right panel showing E-cadherin). Images were taken every 5 min. Time (in minutes) from NEB is indicated in the top left corner. Video is related to Fig. 2 A. Playback speed: 7 frames per second.

**Provided online are two tables. Table S1 shows organoid culturing media. Table S2 shows oligo sequences of gRNAs, primers used for genotyping, and primers used for qPCRs.**

