## [Peer Review File · The Journal of Cell Biology]

Binucleated human hepatocytes arise through late cytokinetic regression during endomitosis M phase

Gabriella Darmasaputra, Cindy Geerlings, Susana Chuva de Sousa Lopes, Hans Clevers, and Matilde Galli

Corresponding Author(s): Matilde Galli, Hubrecht Institute for Developmental Biology and Stem Cell Research

Review Timeline:

Submission Date:	2024-03-04
Editorial Decision:	2024-04-16
Revision Received:	2024-04-24

Monitoring Editor: Daniela Cimini

Scientific Editor: Andrea Marat

Transaction Report:

DOI: <https://doi.org/10.1083/jcb.202403020>

Revision 0

Review #1

1. Evidence, reproducibility and clarity:

Evidence, reproducibility and clarity (Required)

The study provides insights into how polyploidization via endomitosis may arise in human hepatocytes by studying fetal liver cell line-derived organoids. Using live cell imaging and LSM microscopy, binucleation was consistently observed in two independent cell line systems, at frequencies seen in human liver and sensitive to pharmacological inhibition (GSK3i) and genetic manipulation (E2F7 & E2F8 editing). The findings presented are in line with earlier data, largely gathered studying rodents. The data is convincing and robust indicating that these systems can be used to study cause and consequences of polyploidy in human hepatocytes.

While the authors do suggest that they provide a mechanisms how polyploidy is initiated in human hepatocytes undergoing endomitosis, ie. loss of membrane association of membrane-anchoring proteins at the midbody (e.g. Anillin, RacGAP1), I do feel that the data provided is rather descriptive and does not address a particular mechanism that may account for loss of membrane anchoring. As such, the title is making a too strong point, as, in my point of view, it associates with loss of membrane anchorage, but may not drive endomitosis. Whether this is a "passive" process in response to changes in physical forces and tension, or regulated via signalling intermediates to initiate regression of the cleavage furrow is not addressed experimentally (mislocalizing these proteins on a larger scale). Discussion seems warranted.

I do not see the need for additional experiments, as I believe the data is robust and introduces an interesting new model where the role of ploidy can be studied in human hepatocytes ex vivo. However, if the authors wish to extend their studies and document further similarities with pathways engaged in rodents, some E2F7/8 targets relevant for ploidy control such as Anillin or PIDDosome components, or, maybe MDM2 processing for p53 activation, could be tested in wt and E2F mutant cell lines.

A minor suggestion is to clarify the term M-CDK activity in the introduction, as it may not be fully intuitive to all readers; similarly, ploidy reversal is still controversial in the field, but it is stated as a given fact.

2. Significance:

Significance (Required)

Polyploidy at the cellular and nuclear level is a key feature of hepatocytes albeit the physiological significance of the process is not entirely clear. Increased ploidy has been linked to

cancer resistance in the liver, but may pose a threat to hepatocyte survival under conditions of repeated compensatory proliferation cycles. Curiously, during normal regeneration after single surgical intervention liver regeneration is not compromised, even though it may recover faster starting when starting from higher ploidy levels. Mechanistically, most data has been generated studying rodents where it is documented that the proliferation behaviour changes around the time of weaning in mice when hepatocytes start to fail cytokinesis and undergo endomitosis, leading to cellular and nuclear polyploidy. In rodents, insulin signalling / AKT appears involved as is the E2F network and p53, activated by the caspase-2-PIDDosome. The model system introduced here will allow mechanistic studies in human organoids and help to increase our understanding of this process in steady state and under conditions of stress.

3. How much time do you estimate the authors will need to complete the suggested revisions:

Estimated time to Complete Revisions (Required)

(Decision Recommendation)

Less than 1 month

Yes

Review #2

1. Evidence, reproducibility and clarity:

Evidence, reproducibility and clarity (Required)

****Summary:****

Polyploid cells arise within various human tissues by multiple different mechanisms. Here, Darmasaputra et al present a study of one such mechanism, endomitosis, in liver cells using fetal-derived human hepatocyte organoids. In this model, they demonstrate that binucleated cells arise through the late regression of the cytokinetic furrow prior to abscission. They identify a rare event in cytokinetic cells - loss of midbody association with the plasma membrane - that could explain the cytokinesis failure observed in a proportion of these cells. Finally, they show that loss of Wnt signalling increases the number of binucleation events in a manner that depends on E2F7 and E2F8, similar to what has been observed in murine hepatocytes.

****Major comments:****

This is a compelling and well-presented study. The data presented are high quality, the experiments are well described and controlled and the conclusions are convincing. I am particularly impressed by the technical effort that the authors must have put into obtaining high quality live and IF images of dividing cells within organoids and their careful documentation of what are very rare mitotic events. In addition, the manuscript is extremely well written and I found it a pleasure to read.

I do not think that there are additional experiments that are essential to justify the conclusions of the paper. However, I do have suggestions that I think would strengthen this work and increase its significance. As is, the authors present findings in two different areas: the documentation of cytokinesis failure in hepatocyte organoids and the role of Wnt and E2F7/8 on binucleation. It would be really nice if the two parts could be linked. For example, the authors could examine cell divisions in the organoids without Wnt either live or fixed and show that they have a higher proportion of cells undergoing cytokinetic regression or with membrane-midbody attachment defects. Alternatively, they could look at whether the expression levels of key cytokinetic genes are changed in the Wnt and E2F7/8 organoids. As I said, these experiments are not required for or the publication of this work and I will leave it up to the authors to decide if they have the time or capacity to add additional data.

Finally, before publication, the authors should discuss further the mechanisms by which loss of membrane attachment during cytokinesis could occur - there is quite a lot of literature in this area on the role of RacGAP1 and Ect2 in membrane attachment that is not discussed, particularly from the lab of Mark Pentronczki (eg Kotynkova 2026 PMID: 27926870, Lekmotsev PMID: 23235882). It's surprising that the authors haven't mentioned (or looked at) Ect2 at all, especially since Ect2 levels have been shown to control polyploidy in cardiomyocytes (Liu 2019 PMID: 31597755). This at least warrants some discussion.

****Minor comments:****

- Table 1 would be more striking as a graphical representation. I appreciate that the n numbers in the regressed cells means that statistical comparisons is not possible, but some kind of colour coding or graph would make this part clearer
- It's not clear what the difference between Hep-Org 1 and Hep-Org 2 are. Are these from different donors?

2. Significance:

Significance (Required)

This study is an important technical development in that it reports a new system to study in depth cell biology of liver endomitosis in non-transformed and, crucially, human 3D hepatocyte organoids. The findings reported using this system are potentially interesting although they could be further developed if they were mechanistically linked together (see major comments). This work is likely to be highly interesting to scientists studying cell division, cytokinesis and hepatocyte biology. It also has wider implications for liver biology and particularly liver regeneration. Additionally, given the role of polyploidisation in many different tissues, it will likely be of interest to scientists studying polyploidy and endomitosis more generally.

My area of expertise is in cytokinesis and cell division in general, although not specifically in hepatocytes. I am not an expert in organoids.

3. How much time do you estimate the authors will need to complete the suggested revisions:

Estimated time to Complete Revisions (Required)

(Decision Recommendation)

Between 1 and 3 months

Yes

Review #3

1. Evidence, reproducibility and clarity:

Evidence, reproducibility and clarity (Required)

In this manuscript, Darmasaputra and colleagues took advantage of human hepatocyte organoids (Hep-Org) to investigate the formation of binucleated cells that naturally occurs in liver. So far, the mechanism of hepatocyte binucleation has been studied in rodents, where binucleated hepatocytes arise upon weaning through an insulin/akt pathway that inhibits furrow contraction in a fraction of cells (Ref. 21, 22). In addition, it is known that E2F7 and E2F8 downstream of the Wnt signaling repress the expression in mouse hepatocytes of several key cytokinetic proteins (AuroraB, Mklp1, Ect2, Racgap1) and thereby promote binucleation (Ref. 23).

Advances:

As seen *in vivo*, the authors first show that a fraction (5-15%) of cells are binucleated in two independently derived human Hep-Orgs. Live cell imaging reveals that binucleation is not due to furrow ingression defects after anaphase but rather arises from post-furrowing intercellular bridge regression. Fixed data suggest that the cytokinetic midbody formed normally but lost its anchorage to the bridge membrane. Activation of the Wnt signaling resulted in a modest but significant increase in the proportion of binucleated cells (4.5 to <8% or 3 to <6% depending on the subfigures). This increase depended on the presence of E2F7 and E2F8. This study represents the first description of binucleation in a human organoid context.

****Major comments****

1. An outstanding question is whether human Hep-Orgs represent a bona-fide model to study the process of human liver binucleation. The absence of cholangiocytes, vascularization, other cell types and physiological hormones etc. might impact on the mechanism of binucleation, which is the main focus of this study. Since the mechanism of binucleation in human Hep-Orgs appears radically different from what has been reported *in vivo* in rodents, the authors should reproduce the lack of furrow ingression in mouse Hep-Orgs (that they were able to generate in Ref. 44). This could be done in fixed cells as in Fig. 3. Alternatively, they could use live cell imaging and chemical dyes such as SiR-Tubulin and Cell Mask to label microtubules and the plasma membrane, respectively, without the need of creating genome-edited reporter lines.
2. The videos acquired in Fig. 2 contain much more information than presented. The authors should measure the rate of furrow ingression, the extend of spindle elongation, the time of MT severing and the time of furrow/bridge regression after cytokinesis onset. All these parameters are important since spindle elongation and furrow ingression are altered in rodents. Is this also the case in human Hep-orgs? Furthermore, the spindle seems very different (bent bridges) in endomitotic compared to canonical cytokinesis (Fig. 2A). Finally, the authors should provide more time points during the time of furrow regression to better show how this phenomenon occurs. It seems, based on fixed images, that the midbody stays attached to the plasma membrane in an asymmetric manner (i.e. does not fully detach, contrary to what is stated in the text). 3D reconstructions in fixed cells and a further characterization of the movies would clarify this point.
3. DAPI staining is not sensitive enough to detect thin chromatin bridges. To rule out that post-furrowing regression is not merely due to the present of DNA bridges, the authors should confirm their results with LAP2b staining (see PMID 19203582).

4. The authors shows that binucleation results from defective anchorage of the bridge membrane to the midbody, but the molecular mechanism remains elusive and should be further probed. In Fig. 3, there is no obvious changes in the investigated markers. Are the intensities of RACGAP1, Anillin, CIT-K reduced in regressing cells? Are ECT2, activated (phospho) Myosin II, CEP55/ESCRT-III, (activated) AuroraB and MKLP1 normally localized/concentrated? ECT2, AuroraB and MKLP1 are regulated by E2F7/8 (Ref. 23) and AuroraB inactivation after bridge formation leads to late regression (PMID 19203582).
5. The results of Fig. 4F indicate that the increased proportion of binucleated cells upon CHIR99021 removal depends on E2F7/8. Without live cell imaging (or FISH experiments) the authors cannot conclude that the increase in endomitosis is dependent on E2F7/8. A decrease in binucleation could indeed not imply a reduced occurrence of endomitosis. For instance, it is possible that E2F7/8 KO induces the formation of mononucleated 4n cells due to early mitotic failure. This issue should be clarified.
6. Binucleation increases with age both in humans and rodents. Could this feature be mimicked in the human Hep-Org by leaving the organoids longer in culture? (optional but would reinforce the value of the model).

****Minor comments****

1. The results of Table 1 are based on very few fixed cells (3 to 6). The authors should consider increasing the number of regressing cells.
2. Is WNT signaling modified by E2F7/8 mutations? To conclude that "WNT signaling inhibits binucleation in an E2F7/8-dependent manner", the authors should check that E2F7/8 KO does not impair the increase of WNT signaling upon CHIR99021 removal.
3. Please provide movies of the cells presented in Fig. 2A.
4. Removal of CHIR99021 induces major shape changes and lumen formation (rather than "exhibited some morphological changes" as stated). Could the author speculate on this?
5. Fig. 4: Why do the authors use the cell line-1 that has the lowest level of binucleation in this experiment? Would the results be the same in cell line 2? (optional)
6. Would insulin increase the proportion of binucleated cells, as in rodents? (optional)

2. Significance:

Significance (Required)

Strengths and limitations:

The manuscript is well written, easy to follow, and the quality of the data is overall high. A clear strength of this study is the use of state-of-the-art human hepatocyte organoids and genome editing (to generate reporter lines and to KO E2F7/8). This allows the authors to address the mechanism of binucleation in a human context. Interestingly, it revealed both similarities (e.g. E2F7/8 depends for binucleation) and striking mechanistic differences (e.g. post-furrowing regression) between rodent and human systems. The study is rather descriptive -which is fine- but deeper mechanistic insights would strengthen the conclusions of the manuscript. For

instance, "our results identify how human hepatocytes inhibit cell division in endomitosis" appears as an overstatement since the molecular reason of midbody anchorage defects remains elusive.

Audience:

broad, basic research.

Field of expertise:

cell biology of cytokinesis

3. How much time do you estimate the authors will need to complete the suggested revisions:

Estimated time to Complete Revisions (Required)

(Decision Recommendation)

Between 3 and 6 months

Yes

Manuscript number: RC-2023-01965R

Corresponding author(s): Matilde Galli

1. General Statements

In this work we address the mechanism of endomitosis in human hepatocytes using fetal-derived human hepatocyte organoids (Hep-Orgs). Endomitosis is a cell-cycle variation that leads to multinucleated polyploid cells, and is known to occur in the mammalian liver, as well as many other tissues and organs. It is currently unclear how cells switch between canonical and endomitosis cell cycles, and how cells undergoing endomitosis M phase abort cytokinesis to become binucleated. Studies on endomitosis in the liver have been limited to rodents, and although we know that human livers contain many polyploid cells, it is unknown how these binucleated cells arise. Here, we make use of human Hep-Org cultures to provide the first insights into the mechanism of binucleation in human hepatocytes. We find that hepatocytes undergoing endomitosis M phase have normal mitotic timings however they undergo a late cytokinetic regression, leading to binucleated cells. We find that WNT signaling and E2F7/E2F8 determine the proportion of binucleated cells in Hep-Orgs, suggesting that similar mechanisms that control endomitosis in rodents are conserved in human hepatocytes. Our findings demonstrate both differences and similarities between endomitosis in human hepatocytes versus what has been described in rodent hepatocytes, and contribute a valuable system to study how cell-cycle variations evolve during multicellular development.

Overall, the reviewers appreciate our work and agree that our study provides important insights into the mechanism of endomitosis in human hepatocytes. The reviewers gave helpful advice to improve our manuscript, suggesting some new experiments, analyses and textual changes. We have now performed and incorporated the majority of these points in the revised version.

We thank the reviewers for their kind works and helpful insights and suggestions. Below, we have pasted the reviews (in black), with our responses (in blue):

Reviewer #1 (Evidence, reproducibility and clarity (Required)):

The study provides insights into how polyploidization via endomitosis may arise in human hepatocytes by studying fetal liver cell line-derived organoids. Using live cell imaging and LSM microscopy, binucleation was consistently observed in two independent cell line systems, at frequencies seen in human liver and sensitive to pharmacological inhibition (GSK3i) and genetic manipulation (E2F7 & E2F8 editing). The findings presented are in line with earlier data, largely gathered studying rodents. The data is convincing and robust indicating that these systems can be used to study cause and consequences of polyploidy in human hepatocytes.

1. While the authors do suggest that they provide a mechanisms how polyploidy is initiated in human hepatocytes undergoing endomitosis, ie. loss of membrane association of membrane-anchoring proteins at the midbody (e.g. Anillin, RacGAP1), I do feel that the data provided is rather descriptive and does not address a particular mechanism that may account for loss of membrane anchoring. As such, the title is making a too strong point, as, in my point of view, it associates with loss of membrane anchorage, but may not drive endomitosis. Whether this is a "passive" process in response to changes in physical forces and tension, or regulated via signalling intermediates to initiate regression of the cleavage furrow is not addressed experimentally (mislocalizing these proteins on a larger scale). Discussion seems warranted.

We agree with the reviewer that our mechanistic insights into the molecular mechanisms of endomitosis are limited, and we cannot currently prove that the loss of membrane-anchoring drives endomitosis. We have therefore toned down this conclusion and changed the title to "Binucleated human hepatocytes arise through late cytokinetic regression during endomitosis M phase". Furthermore, we have expanded the Discussion to reflect on the gaps in knowledge and speculate about possible molecular mechanisms of endomitosis, see pages 12-16 (in particular, lines 404-423, lines 433-443, and 445-472).

2. I do not see the need for additional experiments, as I believe the data is robust and introduces an interesting new model where the role of ploidy can be studied in human hepatocytes *ex vivo*. However, if the authors wish to extend their studies and document further similarities with pathways engaged in rodents, some E2F7/8 targets relevant for ploidy control such as Anillin or PIDDosome components, or, maybe MDM2 processing for p53 activation, could be tested in wt and E2F mutant cell lines.

Unfortunately, we have not been able to look at E2F7/8 targets and their expression in E2F mutant Hep-Orgs. We performed qPCRs for some cytokinesis regulators such as *Ect2*, *RacGap1* and *Mklp1* in Hep-Orgs, however these genes are so lowly expressed that we can hardly detect them. This is likely because these transcripts are only expressed in a short period of the cell cycle during S/G2 phase, whereas the vast majority of cells in Hep-Orgs are in G1. Therefore, differences in gene expression are very difficult (if not impossible) to detect by qPCR. We also tried to perform single molecule FISH on Hep-Orgs, which would allow us to quantify lowly expressed transcripts in single cells, however despite that the smFISH stainings work well on cholangiocyte organoids and intestinal organoids, we could not get good signals in Hep-Orgs. Taken together, we are unable at this point to look into downstream targets of E2F7/8.

3. A minor suggestion is to clarify the term M-CDK activity in the introduction, as it may not be fully intuitive to all readers; similarly, ploidy reversal is still controversial in the field, but it is stated as a given fact.

Full Revision

Thank you for these suggestions, we have clarified the term M-CDK on page 3, lines 60-61, and have rephrased the sentence on ploidy reversal on page 3, lines 81-82.

Reviewer #1 (Significance (Required)):

Polyploidy at the cellular and nuclear level is a key feature of hepatocytes albeit the physiological significance of the process is not entirely clear. Increased ploidy has been linked to cancer resistance in the liver, but may pose a threat to hepatocyte survival under conditions of repeated compensatory proliferation cycles. Curiously, during normal regeneration after single surgical intervention liver regeneration is not compromised, even though it may recover faster starting when starting from higher ploidy levels. Mechanistically, most data has been generated studying rodents where it is documented that the proliferation behaviour changes around the time of weaning in mice when hepatocytes start to fail cytokinesis and undergo endomitosis, leading to cellular and nuclear polyploidy. In rodents, insulin signalling / AKT appears involved as is the E2F network and p53, activated by the caspase-2-PIDDosome. The model system introduced here will allow mechanistic studies in human organoids and help to increase our understanding of this process in steady state and under conditions of stress.

Reviewer #2 (Evidence, reproducibility and clarity (Required)):

Summary:

Polyploid cells arise within various human tissues by multiple different mechanisms. Here, Darmasaputra et al present a study of one such mechanism, endomitosis, in liver cells using fetal-derived human hepatocyte organoids. In this model, they demonstrate that binucleated cells arise through the late regression of the cytokinetic furrow prior to abscission. They identify a rare event in cytokinetic cells - loss of midbody association with the plasma membrane - that could explain the cytokinesis failure observed in a proportion of these cells. Finally, they show that loss of Wnt signalling increases the number of binucleation events in a manner that depends on E2F7 and E2F8, similar to what has been observed in murine hepatocytes.

Major comments:

This is a compelling and well-presented study. The data presented are high quality, the experiments are well described and controlled and the conclusions are convincing. I am particularly impressed by the technical effort that the authors must have put into obtaining high quality live and IF images of dividing cells within organoids and their careful documentation of what are very rare mitotic events. In addition, the manuscript is extremely well written and I found it a pleasure to read.

1. I do not think that there are additional experiments that are essential to justify the conclusions of the paper. However, I do have suggestions that I think would strengthen this

work and increase its significance. As is, the authors present findings in two different areas: the documentation of cytokinesis failure in hepatocyte organoids and the role of Wnt and E2F7/8 on binucleation. It would be really nice if the two parts could be linked. For example, the authors could examine cell divisions in the organoids without Wnt either live or fixed and show that they have a higher proportion of cells undergoing cytokinetic regression or with membrane-midbody attachment defects. Alternatively, they could look at whether the expression levels of key cytokinetic genes are changed in the Wnt and E2F7/8 organoids. As I said, these experiments are not required for or the publication of this work and I will leave it up to the authors to decide if they have the time or capacity to add additional data.

We thank the reviewer for this suggestion. Unfortunately, despite substantial effort, we have been unable to perform successful live imaging of Hep-Orgs under CHIR99021 removal conditions: these organoids become very sensitive to live imaging and they also proliferate very slowly. We have tried to look at the expression of cytokinetic genes by qPCR, however these experiments were inconclusive (see also our response to reviewer #1, point 2). Thus, we cannot rule out that the increase in binucleation that we see upon CHIR99021 removal is not due to increased endomitosis, but rather occurs independently, for example by an increased survival rate of binucleated cells upon WNT removal. We have now discussed this issue and explained the limitations of our study in the discussion, pages 14-15, lines 451-460.

2. Finally, before publication, the authors should discuss further the mechanisms by which loss of membrane attachment during cytokinesis could occur - there is quite a lot of literature in this area on the role of RacGAP1 and Ect2 in membrane attachment that is not discussed, particularly from the lab of Mark Pentronczki (eg Kotynkova 2026 PMID: 27926870, Lekmotsev PMID: 23235882). It's surprising that the authors haven't mentioned (or looked at) Ect2 at all, especially since Ect2 levels have been shown to control polyploidy in cardiomyocytes (Liu 2019 PMID: 31597755). This at least warrants some discussion.

We thank the reviewer for pointing us to these articles. We have elaborated the discussion to include the work on rodent and human cardiomyocytes, and to explain why we think that there is no defect in ECT2 and RhoA signaling in human hepatocytes undergoing endomitosis, see pages 13-14, 404-423 and 433-443.

Minor comments:

3. Table 1 would be more striking as a graphical representation. I appreciate that the n numbers in the regressed cells means that statistical comparisons is not possible, but some kind of colour coding or graph would make this part clearer

We agree that Table 1 was difficult to read – we now show the data schematically in a new figure, Fig.4.

4. It's not clear what the difference between Hep-Org 1 and Hep-Org 2 are. Are these from different donors?

Indeed Hep-Org1 and Hep-Org2 are from different donors. We have clarified this in the text, see page 5, lines 131-133.

Reviewer #2 (Significance (Required)):

This study is an important technical development in that it reports a new system to study in depth cell biology of liver endomitosis in non-transformed and, crucially, human 3D hepatocyte organoids. The findings reported using this system are potentially interesting although they could be further developed if they were mechanistically linked together (see major comments). This work is likely to be highly interesting to scientists studying cell division, cytokinesis and hepatocyte biology. It also has wider implications for liver biology and particularly liver regeneration. Additionally, given the role of polyploidisation in many different tissues, it will likely be of interest to scientists studying polyploidy and endomitosis more generally.

My area of expertise is in cytokinesis and cell division in general, although not specifically in hepatocytes. I am not an expert in organoids.

Reviewer #3 (Evidence, reproducibility and clarity (Required)):

In this manuscript, Darmasaputra and colleagues took advantage of human hepatocyte organoids (Hep-Org) to investigate the formation of binucleated cells that naturally occurs in liver. So far, the mechanism of hepatocyte binucleation has been studied in rodents, where binucleated hepatocytes arise upon weaning through an insulin/akt pathway that inhibits furrow contraction in a fraction of cells (Ref. 21, 22). In addition, it is known that E2F7 and E2F8 downstream of the Wnt signaling repress the expression in mouse hepatocytes of several key cytokinetic proteins (AuroraB, Mklp1, Ect2, Racgap1) and thereby promote binucleation (Ref. 23).

Advances:

As seen in vivo, the authors first show that a fraction (5-15%) of cells are binucleated in two independently derived human Hep-Orgs. Live cell imaging reveals that binucleation is not due to furrow ingression defects after anaphase but rather arises from post-furrowing intercellular bridge regression. Fixed data suggest that the cytokinetic midbody formed normally but lost its anchorage to the bridge membrane. Activation of the Wnt signaling resulted in a modest but significant increase in the proportion of binucleated cells (4.5 to <8% or 3 to <6% depending on

the subfigures). This increase depended on the presence of E2F7 and E2F8. This study represents the first description of binucleation in a human organoid context.

Major comments

1. An outstanding question is whether human Hep-Orgs represent a bona-fide model to study the process of human liver binucleation. The absence of cholangiocytes, vascularization, other cell types and physiological hormones etc. might impact on the mechanism of binucleation, which is the main focus of this study. Since the mechanism of binucleation in human Hep-Orgs appears radically different from what has been reported in vivo in rodents, the authors should reproduce the lack of furrow ingression in mouse Hep-Orgs (that they were able to generate in Ref. 44). This could be done in fixed cells as in Fig. 3. Alternatively, they could use live cell imaging and chemical dyes such as SiR-Tubulin and Cell Mask to label microtubules and the plasma membrane, respectively, without the need of creating genome-edited reporter lines.

The mechanism of endomitosis that we observe in human hepatocyte organoids is indeed different from what has been observed in mouse hepatocytes. Unfortunately, mouse Hep-Orgs are more difficult to generate as they require a two-step perfusion protocol from live mice (described in Hu et al., 2018). Additionally, mouse Hep-Orgs do not survive freezing, so to be able to perform the suggested experiments, we would need to generate new mouse Hep-Org lines. As our collaborators are currently not performing any experiments with mouse livers, we would need to request an ethical permit to generate these organoids, which would take several months. We have seriously considered this option, however due to the substantial investment in time and resources, we feel these experiments would be more suited for a follow-up study.

To nonetheless better clarify the differences between what has been observed in rodents, and what we see in the Hep-Orgs, we have added a paragraph in the discussion, see pages 14-15 lines 433-460.

2. The videos acquired in Fig. 2 contain much more information than presented. The authors should measure the rate of furrow ingression, the extend of spindle elongation, the time of MT severing and the time of furrow/bridge regression after cytokinesis onset. All these parameters are important since spindle elongation and furrow ingression are altered in rodents. Is this also the case in human Hep-orgs? Furthermore, the spindle seems very different (bent bridges) in endomitotic compared to canonical cytokinesis (Fig. 2A). Finally, the authors should provide more time points during the time of furrow regression to better show how this phenomenon occurs. It seems, based on fixed images, that the midbody stays attached to the plasma membrane in an asymmetric manner (i.e. does not fully detach, contrary to what is stated in the text). 3D reconstructions in fixed cells and a further characterization of the movies would clarify this point.

We thank the reviewer for this suggestion. Although there are some technical limitations that pose some restrictions (explained below), we have extended our analyses where possible. In our live

imaging, we use 5-minute time intervals with 4 μm z-slices, which allows a delicate balance between having enough frames in M phase, and imaging for at least 48 hours, which is required to catch enough divisions. We are unable to image with smaller time intervals or smaller z-slices, as this leads to phototoxicity. Nonetheless, using these settings, we can get an indication of the rate of furrow ingression, time of severing and the time of furrow regression:

- We find that the time of furrowing onset and the rate of furrow ingression is very similar between canonical M phases and endomitosis M phases: we have now added this data in the results section, page 7, lines 192-199 and Fig. 2D.
- The time of cytokinetic regression is more variable between endomitoses events, and can range between 30 minutes and 2,5 hours. We have also added this information to page 7, lines 199-202 and Fig. 2E
- The time of MT severing is similar between endomitosis M phases, as we show in Fig. 2C
- Unfortunately, we cannot accurately measure the extent of spindle elongation, as the divisions occur in 3D and our Z resolution is not good enough. Regarding the observation that the spindle looks different in the endomitosis example in Fig. 2A: we have quantified how often we observe bent midzones in endomitosis versus canonical M phases, and this occurs in 60% of canonical (n=12/20) and 83% of endomitosis M phases (n=15/18). We have now added this information in the results section, page 6, lines 1862-185.
- We have quantified how often we see the midbody remaining attached to one side of the plasma membrane versus fully detaching: we find that in 6 out of 9 late stage endomitotic regressions, the membrane is detached from both sides, and in 3 out of 9, it remains attached to one side. We have added this information to the results section, page 8 line 249-251.

3. DAPI staining is not sensitive enough to detect thin chromatin bridges. To rule out that post-furrowing regression is not merely due to the present of DNA bridges, the authors should confirm their results with LAP2b staining (see PMID 19203582).

To exclude the presence of ultrafine DNA bridges during anaphase, we have performed a staining for RIF1, a factor that localizes to ultrafine DNA bridges in anaphase and is required for their resolution (Hengeveld et al, 2015, PMID: 26256213). In early anaphase, we find many RIF1-positive thread-like structures, as has been described before in other non-transformed and non-stressed cells. However, in late anaphase and telophase, we never observe these fibers (n=57/57), suggesting that they are fully resolved and are not the cause of cytokinetic regression. We have added this data to the results section, see page 8, lines 226-234, and Fig. S1.

4. The authors shows that binucleation results from defective anchorage of the bridge membrane to the midbody, but the molecular mechanism remains elusive and should be further probed. In Fig. 3, there is no obvious changes in the investigated markers. Are the intensities of RACGAP1, Anillin, CIT-K reduced in regressing cells? Are ECT2, activated (phospho) Myosin II, CEP55/ESCRT-III, (activated) AuroraB and MKLP1 normally localized/concentrated? ECT2,

AuroraB and MKLP1 are regulated by E2F7/8 (Ref. 23) and AuroraB inactivation after bridge formation leads to late regression (PMID 19203582).

We agree with the reviewer that the molecular mechanism by which midbodies lose their attachment to the membrane is currently unclear. We do not see any clear differences in the intensities of RACGAP1, Anillin, or CIT-K in cells undergoing endomitotic regression. We also do not expect large differences in localization or abundancies of ECT2, AuroraB or MKLP1, because if this were the case, you would expect differences in early cytokinesis in endomitosis, such as a delay or a slower rate of furrow ingression. We did perform additional IF experiments to investigate the localization of SEPT9, a septin that is expressed in human hepatocytes and that has essential functions in membrane anchorage during cytokinesis. Although we find that SEPT9 exhibits more variable localizations than RACGAP1, Anillin, and CIT-K, we find that in the majority of endomitotic regressions, it is also absent from the regressed membrane (n=5/7 cells). We have added this data to the results section on page 9, and in the figures Fig. 3C and Fig.4C.

5. The results of Fig. 4F indicate that the increased proportion of binucleated cells upon CHIR99021 removal depends on E2F7/8. Without live cell imaging (or FISH experiments) the authors cannot conclude that the increase in endomitosis is dependent on E2F7/8. A decrease in binucleation could indeed not imply a reduced occurrence of endomitosis. For instance, it is possible that E2F7/8 KO induces the formation of mononucleated 4n cells due to early mitotic failure. This issue should be clarified.

The reviewer raises an important point. Unfortunately, we were unable to generate E2F7/8 KO lines containing fluorescent nuclear and membrane markers, which would allow us to perform live-imaging and confirm that these organoids perform less endomitosis. As an alternative, we tried to use SiR-Tubulin dyes for live imaging, but even at very low concentrations these dyes are toxic for the organoids. To exclude the possibility that E2F7/8 KO induces the formation of mononucleated 4n cells, we have measured the DNA content in wildtype, E2F7 and E2F8 lines, and found that the distribution of ploidies is very similar between these lines, both in normal growth conditions as well as upon removal of CHIR99021 (see the new supplemental figure, Fig. S3). We thus think it is unlikely that E2F7/8 KO induces the formation of mononucleated 4n, however it remains possible that the differences in percentage of binucleated cells arise independently of endomitosis. We have now toned down our conclusions on the function of WNT signaling and E2F7/8 in endomitosis, and discussed alternative explanations for our findings in the discussion, see page 14, lines 451-460.

6. Binucleation increases with age both in humans and rodents. Could this feature be mimicked in the human Hep-Org by leaving the organoids longer in culture? (optional but would reinforce the value of the model).

We do not see an increase in binucleation percentages in organoids that are kept longer in cultures, and we have now also tested the effect of growing the organoids in a “differentiation medium”, which was previously described to give rise to more mature hepatocyte gene expression

(Hu et al. 2018), however we see no significant differences in the percentages of binucleated cells per organoid. We have now included this data, as well as our analyses of the effect of insulin in the growth medium (see our response to point 12 below) in the results section on page 11 lines 341-353 and we further discuss this point in the Discussion, pages 12-13, lines 389-397.

Minor comments

7. The results of Table 1 are based on very few fixed cells (3 to 6). The authors should consider increasing the number of regressing cells.

We are aware that the number of endomitotic regressions is very low. Unfortunately, it is extremely challenging to catch these events by IF: cells in Hep-Orgs cycle very slowly (they divide once every ± 50 hours), and thus very few cells are in M phase at any given moment (only 1 or 2 cells per organoid) – the chance that this cell is then also in telophase is even lower, and then only $\pm 5\%$ of the telophase are actually undergoing endomitosis. Due to technical limitations of the organoid IF staining protocol, it is not trivial to scale up these experiments, making it very difficult to find more than 3-5 endomitotic regressions per condition. Despite the low numbers of endomitotic regressions that we have identified, we find that RacGAP1, Anillin and CIT-K localize in a very similar manner in cells undergoing endomitosis. For SEPT9, we see a little bit more variation in the localizations, but also here the majority of cells in undergoing endomitosis have lost SEPT9 membrane association on the regressed membrane (see Fig. 4C).

8. Is WNT signaling modified by E2F7/8 mutations? To conclude that "WNT signaling inhibits binucleation in an E2F7/8-dependent manner", the authors should check that E2F7/8 KO does not impair the increase of WNT signaling upon CHIR99021 removal.

We had not thought of this option, but it is indeed possible that E2F7/8 influences the ability of cells to respond to CHIR99021 removal. WNT regulators are not known to be targets of E2F7 or E2F8 in mice (see PMIDs: 22180533, 18194653, and 23064264), however as we have not analyzed the gene expression changes in E2F7 or E2F8 mutant organoids, we cannot exclude the possibility that CHIR99021 has different effects in E2F7/E2F8 knock-out cells. We now discuss this possibility in the discussion, page 15, lines 459-460.

9. Please provide movies of the cells presented in Fig. 2A.

We have included movies of these cells, see Supplemental Movie 1 and Supplemental Movie 2.

10. Removal of CHIR99021 induces major shape changes and lumen formation (rather than "exhibited some morphological changes" as stated). Could the author speculate on this?

WNT signaling is likely important for many aspects of hepatocyte growth and differentiation, and it is possible that upon CHIR99021 removal, Hep-Orgs are starting to differentiate and become more secretory, which would explain why they start forming larger lumens. We now discuss this

Full Revision

in more detail in the final part of the results section, see page 11 lines 341-353, and in the discussion, page 15 lines 462-472.

11. Fig. 4: Why do the authors use the cell line-1 that has the lowest level of binucleation in this experiment? Would the results be the same in cell line 2? (optional)

We perform most experiments in Hep-Org line 1 because this line is easier to maintain in culture, and we have been unable to generate CRISPR knock-outs in Hep-Org line 2.

12. Would insulin increase the proportion of binucleated cells, as in rodents? (optional)

We have tested this, but do not see a difference in the percentage of binucleated cells when we either increase or decrease the concentration of insulin in the growth medium. We have now added this data in the results section, see page 11, lines 347-350 and Fig. 5J.

Reviewer #3 (Significance (Required)):

Strengths and limitations:

The manuscript is well written, easy to follow, and the quality of the data is overall high. A clear strength of this study is the use of state-of-the-art human hepatocyte organoids and genome editing (to generate reporter lines and to KO E2F7/8). This allows the authors to address the mechanism of binucleation in a human context. Interestingly, it revealed both similarities (e.g. E2F7/8 depends for binucleation) and striking mechanistic differences (e.g. post-furrowing regression) between rodent and human systems. The study is rather descriptive -which is fine- but deeper mechanistic insights would strengthen the conclusions of the manuscript. For instance, "our results identify how human hepatocytes inhibit cell division in endomitosis" appears as an overstatement since the molecular reason of midbody anchorage defects remains elusive.

We thank the reviewer for their kind words. Unfortunately, we have been unable to gain deeper mechanistic insights into the molecular mechanism of membrane regression in endomitosis. We have therefore toned down our conclusions, see the new concluding sentence in the abstract, page 2, lines 35-36.

Audience:
broad, basic research.

Field of expertise:
cell biology of cytokinesis

April 16, 2024

RE: JCB Manuscript #202403020T

Dr. Matilde Galli
Hubrecht Institute for Developmental Biology and Stem Cell Research
Uppsalalaan 8
Utrecht 3584CT
Netherlands

Dear Dr. Galli:

Thank you for submitting your revised manuscript entitled "Binucleated human hepatocytes arise through late cytokinetic regression during endomitosis M phase". Your study has now been assessed by the original reviewers from Review Commons, whose comments are appended below. We would be happy to publish your paper in JCB pending final revisions necessary to meet our formatting guidelines (see details below). In your final revision, please be sure to address the remaining reviewer concerns with text and figure edits as requested. In particular, the issue of E2F7/8 targets raised by reviewer 1 needs to be discussed, however additional experimental data is not required.

A. MANUSCRIPT ORGANIZATION AND FORMATTING:

- 1) Text limits: Character count for Articles is < 40,000, not including spaces. Count includes abstract, introduction, results, discussion, and acknowledgments. Count does not include title page, figure legends, materials and methods, references, tables, or supplemental legends.
- 2) Figures limits: Articles may have up to 10 main text figures.
- 3) Figure formatting: Scale bars must be present on all microscopy images, including inset magnifications (you may alternatively indicate the diameter of the inset). Molecular weight or nucleic acid size markers must be included on all gel electrophoresis.
- 4) Statistical analysis: Error bars on graphic representations of numerical data must be clearly described in the figure legend. The number of independent data points (n) represented in a graph must be indicated in the legend. Statistical methods should be explained in full in the materials and methods. For figures presenting pooled data the statistical measure should be defined in the figure legends. Please also be sure to indicate the statistical tests used in each of your experiments (either in the figure legend itself or in a separate methods section) as well as the parameters of the test (for example, if you ran a t-test, please indicate if it was one- or two-sided, etc.). Also, if you used parametric tests, please indicate if the data distribution was tested for normality (and if so, how). If not, you must state something to the effect that "Data distribution was assumed to be normal but this was not formally tested."
- 5) Abstract and title: The abstract should be no longer than 160 words and should communicate the significance of the paper for a general audience. The title should be less than 100 characters including spaces. Make the title concise but accessible to a general readership.

* While we agree with reviewer 3 that the limitations need to be clearly outlined, to keep the title concise you must emphasize the limitations of organoids in the abstract as well as in the discussion, instead of the title.
- 6) Materials and methods: Should be comprehensive and not simply reference a previous publication for details on how an experiment was performed. Please provide full descriptions in the text for readers who may not have access to referenced manuscripts.
- 7) All antibodies, cell lines, animals, and tools used in the manuscript should be described in full, including accession numbers for materials available in a public repository such as the Resource Identification Portal. Please be sure to provide the sequences for all of your primers/oligos and RNAi constructs in the materials and methods. You must also indicate in the methods the source, species, and catalog numbers (where appropriate) for all of your antibodies. Please also indicate the acquisition and quantification methods for immunoblotting/western blots.

8) Microscope image acquisition: The following information must be provided about the acquisition and processing of images:

- Make and model of microscope
- Type, magnification, and numerical aperture of the objective lenses
- Temperature
- Imaging medium
- Fluorochromes
- Camera make and model
- Acquisition software
- Any software used for image processing subsequent to data acquisition. Please include details and types of operations involved (e.g., type of deconvolution, 3D reconstitutions, surface or volume rendering, gamma adjustments, etc.).

10) Supplemental materials: There are strict limits on the allowable amount of supplemental data. Articles may have up to 5 supplemental figures. Please also note that tables, like figures, should be provided as individual, editable files. A summary of all supplemental material should appear at the end of the Materials and methods section.

13) ORCID IDs: ORCID IDs are unique identifiers allowing researchers to create a record of their various scholarly contributions in a single place. Please note that ORCID IDs are now *required* for all authors. At resubmission of your final files, please be sure to provide your ORCID ID and those of all co-authors.

Please note that JCB now requires authors to submit Source Data used to generate figures containing gels and Western blots with all revised manuscripts. This Source Data consists of fully uncropped and unprocessed images for each gel/blot displayed in the main and supplemental figures. Since your paper includes cropped gel and/or blot images, please be sure to provide one Source Data file for each figure that contains gels and/or blots along with your revised manuscript files. File names for Source Data figures should be alphanumeric without any spaces or special characters (i.e., SourceDataF#, where F# refers to the associated main figure number or SourceDataFS# for those associated with Supplementary figures). The lanes of the gels/blots should be labeled as they are in the associated figure, the place where cropping was applied should be marked (with a box), and molecular weight/size standards should be labeled wherever possible.

Journal of Cell Biology now requires a data availability statement for all research article submissions. These statements will be published in the article directly above the Acknowledgments. The statement should address all data underlying the research presented in the manuscript. Please visit the JCB instructions for authors for guidelines and examples of statements at (<https://rupress.org/jcb/pages/editorial-policies#data-availability-statement>).

B. FINAL FILES:

-- High-resolution figure and MP4 video files: See our detailed guidelines for preparing your production-ready images,

<https://jcb.rupress.org/fig-vid-guidelines>.

Thank you for your attention to these final processing requirements. Please revise and format the manuscript and upload materials within 7 days. If you need an extension for whatever reason, please let us know and we can work with you to determine a suitable revision period.

Thank you for this interesting contribution, we look forward to publishing your paper in Journal of Cell Biology.

Sincerely,

Daniela Cimini, PhD
Monitoring Editor

Andrea L. Marat, PhD
Senior Scientific Editor

Journal of Cell Biology

Reviewer #1 (Comments to the Authors (Required)):

As commented before, the study is clearly of interest and provides an interesting tool to the community. Otherwise it is largely confirmative suggesting that polyploidization works similar in human hepatocytes and mouse models.

Some minor comments

1. Fig. 3: Ideally, include more or all IFs of regressed cleavage furrows in the figure or supplements
2. Fig. 5: To expand on the mechanism qPCRs on E2F7/8 targets related to cytokinesis in wt and E2F7 and 8 mutant organoid lines o{plus minus} inhibitor withdrawal should still pick up changes in bulk analyses.
3. Fig. 2: why were not all quantifications done in both Hep-Org lines? Is one so much better than the other? If so, please discuss

Regarding mononucleated polyploid cells that may exist also in these organoids, but were not analyzed, one could generate single cell suspensions and perform flow cytometry to measure the DNA content, or simply isolate nuclei for FACS.

Reviewer #2 (Comments to the Authors (Required)):

The authors have fully addressed my comments. I particularly like the new graphical figure 4. I recommend that this high-quality work be published in JCB.

Reviewer #3 (Comments to the Authors (Required)):

The authors provided new data and partially addressed my main concerns. I am happy to recommend publication in JCB when the remaining points are addressed.

1. Major point #1 remains and an important expected result -the increase of BN cells in aged organoids- is not observed. This questions whether this organoid approach really mimics what happens in human liver and whether BN hepatocytes arise from late furrow regression in vivo in Human. I think that this should be stated in a limitation section within the discussion. I also recommend that the title is changed into "Binucleated human hepatocytes arise through late cytokinetic regression during endomitosis M phase in fetal-derived human hepatocyte organoids" or 'Binucleated human hepatocytes arise through late cytokinetic regression in fetal-derived human hepatocyte organoids" (or equivalent wording to clearly state which model has been the basis for this conclusion).

2. Figure 5C panel regressed: It is not clear that this picture shows a cell with a regressed bridge. Please clarify this point and provide a convincing picture. The authors may need to revise the quantifications for SEPT9 accordingly.

Full Revision

Manuscript number: RC-2023-01965R
Corresponding author(s): Matilde Galli

1. General Statements

In this work we address the mechanism of endomitosis in human hepatocytes using fetal-derived human hepatocyte organoids (Hep-Orgs). Endomitosis is a cell-cycle variation that leads to multinucleated polyploid cells, and is known to occur in the mammalian liver, as well as many other tissues and organs. It is currently unclear how cells switch between canonical and endomitosis cell cycles, and how cells undergoing endomitosis M phase abort cytokinesis to become binucleated. Studies on endomitosis in the liver have been limited to rodents, and although we know that human livers contain many polyploid cells, it is unknown how these binucleated cells arise. Here, we make use of human Hep-Org cultures to provide the first insights into the mechanism of binucleation in human hepatocytes. We find that hepatocytes undergoing endomitosis M phase have normal mitotic timings however they undergo a late cytokinetic regression, leading to binucleated cells. We find that WNT signaling and E2F7/E2F8 determine the proportion of binucleated cells in Hep-Orgs, suggesting that similar mechanisms that control endomitosis in rodents are conserved in human hepatocytes. Our findings demonstrate both differences and similarities between endomitosis in human hepatocytes versus what has been described in rodent hepatocytes, and contribute a valuable system to study how cell-cycle variations evolve during multicellular development.

Overall, the reviewers appreciate our work and agree that our study provides important insights into the mechanism of endomitosis in human hepatocytes. The reviewers gave helpful advice to improve our manuscript, suggesting some new experiments, analyses and textual changes. We have now performed and incorporated the majority of these points in the revised version.

We thank the reviewers for their kind works and helpful insights and suggestions. Below, we have pasted the reviews (in black), with our responses (in blue):

Reviewer #1 (Evidence, reproducibility and clarity (Required)):

The study provides insights into how polyploidization via endomitosis may arise in human hepatocytes by studying fetal liver cell line-derived organoids. Using live cell imaging and LSM microscopy, binucleation was consistently observed in two independent cell line systems, at frequencies seen in human liver and sensitive to pharmacological inhibition (GSK3i) and genetic manipulation (E2F7 & E2F8 editing). The findings presented are in line with earlier data, largely gathered studying rodents. The data is convincing and robust indicating that these systems can be used to study cause and consequences of polyploidy in human hepatocytes.

Full Revision

1. While the authors do suggest that they provide a mechanisms how polyploidy is initiated in human hepatocytes undergoing endomitosis, ie. loss of membrane association of membrane-anchoring proteins at the midbody (e.g. Anillin, RacGAP1), I do feel that the data provided is rather descriptive and does not address a particular mechanism that may account for loss of membrane anchoring. As such, the title is making a too strong point, as, in my point of view, it associates with loss of membrane anchorage, but may not drive endomitosis. Whether this is a "passive" process in response to changes in physical forces and tension, or regulated via signalling intermediates to initiate regression of the cleavage furrow is not addressed experimentally (mislocalizing these proteins on a larger scale). Discussion seems warranted.

We agree with the reviewer that our mechanistic insights into the molecular mechanisms of endomitosis are limited, and we cannot currently prove that the loss of membrane-anchoring drives endomitosis. We have therefore toned down this conclusion and changed the title to "Binucleated human hepatocytes arise through late cytokinetic regression during endomitosis M phase". Furthermore, we have expanded the Discussion to reflect on the gaps in knowledge and speculate about possible molecular mechanisms of endomitosis, see pages 12-16 (in particular, lines 404-423, lines 433-443, and 445-472).

2. I do not see the need for additional experiments, as I believe the data is robust and introduces an interesting new model where the role of ploidy can be studied in human hepatocytes ex vivo. However, if the authors wish to extend their studies and document further similarities with pathways engaged in rodents, some E2F7/8 targets relevant for ploidy control such as Anillin or PIDDosome components, or, maybe MDM2 processing for p53 activation, could be tested in wt and E2F mutant cell lines.

Unfortunately, we have not been able to look at E2F7/8 targets and their expression in E2F mutant Hep-Orgs. We performed qPCRs for some cytokinesis regulators such as *Ect2*, *RacGap1* and *Mklp1* in Hep-Orgs, however these genes are so lowly expressed that we can hardly detect them. This is likely because these transcripts are only expressed in a short period of the cell cycle during S/G2 phase, whereas the vast majority of cells in Hep-Orgs are in G1. Therefore, differences in gene expression are very difficult (if not impossible) to detect by qPCR. We also tried to perform single molecule FISH on Hep-Orgs, which would allow us to quantify lowly expressed transcripts in single cells, however despite that the smFISH stainings work well on cholangiocyte organoids and intestinal organoids, we could not get good signals in Hep-Orgs. Taken together, we are unable at this point to look into downstream targets of E2F7/8.

3. A minor suggestion is to clarify the term M-CDK activity in the introduction, as it may not be fully intuitive to all readers; similarly, ploidy reversal is still controversial in the field, but it is stated as a given fact.

Full Revision

Thank you for these suggestions, we have clarified the term M-CDK on page 3, lines 60-61, and have rephrased the sentence on ploidy reversal on page 3, lines 81-82.

Reviewer #1 (Significance (Required)):

Polyploidy at the cellular and nuclear level is a key feature of hepatocytes albeit the physiological significance of the process is not entirely clear. Increased ploidy has been linked to cancer resistance in the liver, but may pose a threat to hepatocyte survival under conditions of repeated compensatory proliferation cycles. Curiously, during normal regeneration after single surgical intervention liver regeneration is not compromised, even though it may recover faster starting when starting from higher ploidy levels. Mechanistically, most data has been generated studying rodents where it is documented that the proliferation behaviour changes around the time of weaning in mice when hepatocytes start to fail cytokinesis and undergo endomitosis, leading to cellular and nuclear polyploidy. In rodents, insulin signalling / AKT appears involved as is the E2F network and p53, activated by the caspase-2-PIDDosome. The model system introduced here will allow mechanistic studies in human organoids and help to increase our understanding of this process in steady state and under conditions of stress.

Reviewer #2 (Evidence, reproducibility and clarity (Required)):

Summary:

Polyploid cells arise within various human tissues by multiple different mechanisms. Here, Darmasaputra et al present a study of one such mechanism, endomitosis, in liver cells using fetal-derived human hepatocyte organoids. In this model, they demonstrate that binucleated cells arise through the late regression of the cytokinetic furrow prior to abscission. They identify a rare event in cytokinetic cells - loss of midbody association with the plasma membrane - that could explain the cytokinesis failure observed in a proportion of these cells. Finally, they show that loss of Wnt signalling increases the number of binucleation events in a manner that depends on E2F7 and E2F8, similar to what has been observed in murine hepatocytes.

Major comments:

This is a compelling and well-presented study. The data presented are high quality, the experiments are well described and controlled and the conclusions are convincing. I am particularly impressed by the technical effort that the authors must have put into obtaining high quality live and IF images of dividing cells within organoids and their careful documentation of what are very rare mitotic events. In addition, the manuscript is extremely well written and I found it a pleasure to read.

1. I do not think that there are additional experiments that are essential to justify the conclusions of the paper. However, I do have suggestions that I think would strengthen this

Full Revision

work and increase its significance. As is, the authors present findings in two different areas: the documentation of cytokinesis failure in hepatocyte organoids and the role of Wnt and E2F7/8 on binucleation. It would be really nice if the two parts could be linked. For example, the authors could examine cell divisions in the organoids without Wnt either live or fixed and show that they have a higher proportion of cells undergoing cytokinetic regression or with membrane-midbody attachment defects. Alternatively, they could look at whether the expression levels of key cytokinetic genes are changed in the Wnt and E2F7/8 organoids. As I said, these experiments are not required for or the publication of this work and I will leave it up to the authors to decide if they have the time or capacity to add additional data.

We thank the reviewer for this suggestion. Unfortunately, despite substantial effort, we have been unable to perform successful live imaging of Hep-Orgs under CHIR99021 removal conditions: these organoids become very sensitive to live imaging and they also proliferate very slowly. We have tried to look at the expression of cytokinetic genes by qPCR, however these experiments were inconclusive (see also our response to reviewer #1, point 2). Thus, we cannot rule out that the increase in binucleation that we see upon CHIR99021 removal is not due to increased endomitosis, but rather occurs independently, for example by an increased survival rate of binucleated cells upon WNT removal. We have now discussed this issue and explained the limitations of our study in the discussion, pages 14-15, lines 451-460.

2. Finally, before publication, the authors should discuss further the mechanisms by which loss of membrane attachment during cytokinesis could occur - there is quite a lot of literature in this area on the role of RacGAP1 and Ect2 in membrane attachment that is not discussed, particularly from the lab of Mark Pentronczki (eg Kotynkova 2026 PMID: 27926870, Lekmotsev PMID: 23235882). It's surprising that the authors haven't mentioned (or looked at) Ect2 at all, especially since Ect2 levels have been shown to control polyploidy in cardiomyocytes (Liu 2019 PMID: 31597755). This at least warrants some discussion.

We thank the reviewer for pointing us to these articles. We have elaborated the discussion to include the work on rodent and human cardiomyocytes, and to explain why we think that there is no defect in ECT2 and RhoA signaling in human hepatocytes undergoing endomitosis, see pages 13-14, 404-423 and 433-443.

Minor comments:

3. Table 1 would be more striking as a graphical representation. I appreciate that the n numbers in the regressed cells means that statistical comparisons is not possible, but some kind of colour coding or graph would make this part clearer

Full Revision

We agree that Table 1 was difficult to read – we now show the data schematically in a new figure, Fig.4.

4. It's not clear what the difference between Hep-Org 1 and Hep-Org 2 are. Are these from different donors?

Indeed Hep-Org1 and Hep-Org2 are from different donors. We have clarified this in the text, see page 5, lines 131-133.

Reviewer #2 (Significance (Required)):

This study is an important technical development in that it reports a new system to study in depth cell biology of liver endomitosis in non-transformed and, crucially, human 3D hepatocyte organoids. The findings reported using this system are potentially interesting although they could be further developed if they were mechanistically linked together (see major comments). This work is likely to be highly interesting to scientists studying cell division, cytokinesis and hepatocyte biology. It also has wider implications for liver biology and particularly liver regeneration. Additionally, given the role of polyploidisation in many different tissues, it will likely be of interest to scientists studying polyploidy and endomitosis more generally.

My area of expertise is in cytokinesis and cell division in general, although not specifically in hepatocytes. I am not an expert in organoids.

Reviewer #3 (Evidence, reproducibility and clarity (Required)):

In this manuscript, Darmasaputra and colleagues took advantage of human hepatocyte organoids (Hep-Org) to investigate the formation of binucleated cells that naturally occurs in liver. So far, the mechanism of hepatocyte binucleation has been studied in rodents, where binucleated hepatocytes arise upon weaning through an insulin/akt pathway that inhibits furrow contraction in a fraction of cells (Ref. 21, 22). In addition, it is known that E2F7 and E2F8 downstream of the Wnt signaling repress the expression in mouse hepatocytes of several key cytokinetic proteins (AuroraB, Mklp1, Ect2, Racgap1) and thereby promote binucleation (Ref. 23).

Advances:

As seen in vivo, the authors first show that a fraction (5-15%) of cells are binucleated in two independently derived human Hep-Orgs. Live cell imaging reveals that binucleation is not due to furrow ingression defects after anaphase but rather arises from post-furrowing intercellular bridge regression. Fixed data suggest that the cytokinetic midbody formed normally but lost its anchorage to the bridge membrane. Activation of the Wnt signaling resulted in a modest but significant increase in the proportion of binucleated cells (4.5 to <8% or 3 to <6% depending on

Full Revision

the subfigures). This increase depended on the presence of E2F7 and E2F8. This study represents the first description of binucleation in a human organoid context.

Major comments

1. An outstanding question is whether human Hep-Orgs represent a bona-fide model to study the process of human liver binucleation. The absence of cholangiocytes, vascularization, other cell types and physiological hormones etc. might impact on the mechanism of binucleation, which is the main focus of this study. Since the mechanism of binucleation in human Hep-Orgs appears radically different from what has been reported in vivo in rodents, the authors should reproduce the lack of furrow ingression in mouse Hep-Orgs (that they were able to generate in Ref. 44). This could be done in fixed cells as in Fig. 3. Alternatively, they could use live cell imaging and chemical dyes such as SiR-Tubulin and Cell Mask to label microtubules and the plasma membrane, respectively, without the need of creating genome-edited reporter lines.

The mechanism of endomitosis that we observe in human hepatocyte organoids is indeed different from what has been observed in mouse hepatocytes. Unfortunately, mouse Hep-Orgs are more difficult to generate as they require a two-step perfusion protocol from live mice (described in Hu et al., 2018). Additionally, mouse Hep-Orgs do not survive freezing, so to be able to perform the suggested experiments, we would need to generate new mouse Hep-Org lines. As our collaborators are currently not performing any experiments with mouse livers, we would need to request an ethical permit to generate these organoids, which would take several months. We have seriously considered this option, however due to the substantial investment in time and resources, we feel these experiments would be more suited for a follow-up study.

To nonetheless better clarify the differences between what has been observed in rodents, and what we see in the Hep-Orgs, we have added a paragraph in the discussion, see pages 14-15 lines 433-460.

2. The videos acquired in Fig. 2 contain much more information than presented. The authors should measure the rate of furrow ingression, the extend of spindle elongation, the time of MT severing and the time of furrow/bridge regression after cytokinesis onset. All these parameters are important since spindle elongation and furrow ingression are altered in rodents. Is this also the case in human Hep-orgs? Furthermore, the spindle seems very different (bent bridges) in endomitotic compared to canonical cytokinesis (Fig. 2A). Finally, the authors should provide more time points during the time of furrow regression to better show how this phenomenon occurs. It seems, based on fixed images, that the midbody stays attached to the plasma membrane in an asymmetric manner (i.e. does not fully detach, contrary to what is stated in the text). 3D reconstructions in fixed cells and a further characterization of the movies would clarify this point.

We thank the reviewer for this suggestion. Although there are some technical limitations that pose some restrictions (explained below), we have extended our analyses where possible. In

Full Revision

our live imaging, we use 5-minute time intervals with 4 μm z-slices, which allows a delicate balance between having enough frames in M phase, and imaging for at least 48 hours, which is required to catch enough divisions. We are unable to image with smaller time intervals or smaller z-slices, as this leads to phototoxicity. Nonetheless, using these settings, we can get an indication of the rate of furrow ingression, time of severing and the time of furrow regression:

- We find that the time of furrowing onset and the rate of furrow ingression is very similar between canonical M phases and endomitosis M phases: we have now added this data in the results section, page 7, lines 192-199 and Fig. 2D.
- The time of cytokinetic regression is more variable between endomitoses events, and can range between 30 minutes and 2,5 hours. We have also added this information to page 7, lines 199-202 and Fig. 2E
- The time of MT severing is similar between endomitosis M phases, as we show in Fig. 2C
- Unfortunately, we cannot accurately measure the extent of spindle elongation, as the divisions occur in 3D and our Z resolution is not good enough. Regarding the observation that the spindle looks different in the endomitosis example in Fig. 2A: we have quantified how often we observe bent midzones in endomitosis versus canonical M phases, and this occurs in 60% of canonical ($n=12/20$) and 83% of endomitosis M phases ($n=15/18$). We have now added this information in the results section, page 6, lines 1862-185.
- We have quantified how often we see the midbody remaining attached to one side of the plasma membrane versus fully detaching: we find that in 6 out of 9 late stage endomitotic regressions, the membrane is detached from both sides, and in 3 out of 9, it remains attached to one side. We have added this information to the results section, page 8 line 249-251.

3. DAPI staining is not sensitive enough to detect thin chromatin bridges. To rule out that post-furrowing regression is not merely due to the present of DNA bridges, the authors should confirm their results with LAP2b staining (see PMID 19203582).

To exclude the presence of ultrafine DNA bridges during anaphase, we have performed a staining for RIF1, a factor that localizes to ultrafine DNA bridges in anaphase and is required for their resolution (Hengeveld et al, 2015, PMID: 26256213). In early anaphase, we find many RIF1-positive thread-like structures, as has been described before in other non-transformed and non-stressed cells. However, in late anaphase and telophase, we never observe these fibers ($n=57/57$), suggesting that they are fully resolved and are not the cause of cytokinetic regression. We have added this data to the results section, see page 8, lines 226-234, and Fig. S1.

4. The authors shows that binucleation results from defective anchorage of the bridge membrane to the midbody, but the molecular mechanism remains elusive and should be further probed. In Fig. 3, there is no obvious changes in the investigated markers. Are the intensities of

Full Revision

RACGAP1, Anillin, CIT-K reduced in regressing cells? Are ECT2, activated (phospho) Myosin II, CEP55/ESCRT-III, (activated) AuroraB and MKLP1 normally localized/concentrated? ECT2, AuroraB and MKLP1 are regulated by E2F7/8 (Ref. 23) and AuroraB inactivation after bridge formation leads to late regression (PMID 19203582).

We agree with the reviewer that the molecular mechanism by which midbodies lose their attachment to the membrane is currently unclear. We do not see any clear differences in the intensities of RACGAP1, Anillin, or CIT-K in cells undergoing endomitotic regression. We also do not expect large differences in localization or abundancies of ECT2, AuroraB or MKLP1, because if this were the case, you would expect differences in early cytokinesis in endomitosis, such as a delay or a slower rate of furrow ingression. We did perform additional IF experiments to investigate the localization of SEPT9, a septin that is expressed in human hepatocytes and that has essential functions in membrane anchorage during cytokinesis. Although we find that SEPT9 exhibits more variable localizations than RACGAP1, Anillin, and CIT-K, we find that in the majority of endomitotic regressions, it is also absent from the regressed membrane (n=5/7 cells). We have added this data to the results section on page 9, and in the figures Fig. 3C and Fig.4C.

5. The results of Fig. 4F indicate that the increased proportion of binucleated cells upon CHIR99021 removal depends on E2F7/8. Without live cell imaging (or FISH experiments) the authors cannot conclude that the increase in endomitosis is dependent on E2F7/8. A decrease in binucleation could indeed not imply a reduced occurrence of endomitosis. For instance, it is possible that E2F7/8 KO induces the formation of mononucleated 4n cells due to early mitotic failure. This issue should be clarified.

The reviewer raises an important point. Unfortunately, we were unable to generate E2F7/8 KO lines containing fluorescent nuclear and membrane markers, which would allow us to perform live-imaging and confirm that these organoids perform less endomitosis. As an alternative, we tried to use SiR-Tubulin dyes for live imaging, but even at very low concentrations these dyes are toxic for the organoids. To exclude the possibility that E2F7/8 KO induces the formation of mononucleated 4n cells, we have measured the DNA content in wildtype, E2F7 and E2F8 lines, and found that the distribution of ploidies is very similar between these lines, both in normal growth conditions as well as upon removal of CHIR99021 (see the new supplemental figure, Fig. S3). We thus think it is unlikely that E2F7/8 KO induces the formation of mononucleated 4n, however it remains possible that the differences in percentage of binucleated cells arise independently of endomitosis. We have now toned down our conclusions on the function of WNT signaling and E2F7/8 in endomitosis, and discussed alternative explanations for our findings in the discussion, see page 14, lines 451-460.

6. Binucleation increases with age both in humans and rodents. Could this feature be mimicked in the human Hep-Org by leaving the organoids longer in culture? (optional but would reinforce the value of the model).

Full Revision

We do not see an increase in binucleation percentages in organoids that are kept longer in cultures, and we have now also tested the effect of growing the organoids in a “differentiation medium”, which was previously described to give rise to more mature hepatocyte gene expression (Hu et al. 2018), however we see no significant differences in the percentages of binucleated cells per organoid. We have now included this data, as well as our analyses of the effect of insulin in the growth medium (see our response to point 12 below) in the results section on page 11 lines 341-353 and we further discuss this point in the Discussion, pages 12-13, lines 389-397.

Minor comments

7. The results of Table 1 are based on very few fixed cells (3 to 6). The authors should consider increasing the number of regressing cells.

We are aware that the number of endomitotic regressions is very low. Unfortunately, it is extremely challenging to catch these events by IF: cells in Hep-Orgs cycle very slowly (they divide once every ± 50 hours), and thus very few cells are in M phase at any given moment (only 1 or 2 cells per organoid) – the chance that this cell is then also in telophase is even lower, and then only $\pm 5\%$ of the telophase are actually undergoing endomitosis. Due to technical limitations of the organoid IF staining protocol, it is not trivial to scale up these experiments, making it very difficult to find more than 3-5 endomitotic regressions per condition. Despite the low numbers of endomitotic regressions that we have identified, we find that RacGAP1, Anillin and CIT-K localize in a very similar manner in cells undergoing endomitosis. For SEPT9, we see a little bit more variation in the localizations, but also here the majority of cells in undergoing endomitosis have lost SEPT9 membrane association on the regressed membrane (see Fig. 4C).

8. Is WNT signaling modified by E2F7/8 mutations? To conclude that "WNT signaling inhibits binucleation in an E2F7/8-dependent manner", the authors should check that E2F7/8 KO does not impair the increase of WNT signaling upon CHIR99021 removal.

We had not thought of this option, but it is indeed possible that E2F7/8 influences the ability of cells to respond to CHIR99021 removal. WNT regulators are not known to be targets of E2F7 or E2F8 in mice (see PMIDs: 22180533, 18194653, and 23064264), however as we have not analyzed the gene expression changes in E2F7 or E2F8 mutant organoids, we cannot exclude the possibility that CHIR99021 has different effects in E2F7/E2F8 knock-out cells. We now discuss this possibility in the discussion, page 15, lines 459-460.

9. Please provide movies of the cells presented in Fig. 2A.

We have included movies of these cells, see Supplemental Movie 1 and Supplemental Movie 2.

Full Revision

10. Removal of CHIR99021 induces major shape changes and lumen formation (rather than "exhibited some morphological changes" as stated). Could the author speculate on this?

WNT signaling is likely important for many aspects of hepatocyte growth and differentiation, and it is possible that upon CHIR99021 removal, Hep-Orgs are starting to differentiate and become more secretory, which would explain why they start forming larger lumens. We now discuss this in more detail in the final part of the results section, see page 11 lines 341-353, and in the discussion, page 15 lines 462-472.

11. Fig. 4: Why do the authors use the cell line-1 that has the lowest level of binucleation in this experiment? Would the results be the same in cell line 2? (optional)

We perform most experiments in Hep-Org line 1 because this line is easier to maintain in culture, and we have been unable to generate CRISPR knock-outs in Hep-Org line 2.

12. Would insulin increase the proportion of binucleated cells, as in rodents? (optional)

We have tested this, but do not see a difference in the percentage of binucleated cells when we either increase or decrease the concentration of insulin in the growth medium. We have now added this data in the results section, see page 11, lines 347-350 and Fig. 5J.

Reviewer #3 (Significance (Required)):

Strengths and limitations:

The manuscript is well written, easy to follow, and the quality of the data is overall high. A clear strength of this study is the use of state-of-the-art human hepatocyte organoids and genome editing (to generate reporter lines and to KO E2F7/8). This allows the authors to address the mechanism of binucleation in a human context. Interestingly, it revealed both similarities (e.g. E2F7/8 depends for binucleation) and striking mechanistic differences (e.g. post-furrowing regression) between rodent and human systems. The study is rather descriptive -which is fine- but deeper mechanistic insights would strengthen the conclusions of the manuscript. For instance, "our results identify how human hepatocytes inhibit cell division in endomitosis" appears as an overstatement since the molecular reason of midbody anchorage defects remains elusive.

We thank the reviewer for their kind words. Unfortunately, we have been unable to gain deeper mechanistic insights into the molecular mechanism of membrane regression in endomitosis. We have therefore toned down our conclusions, see the new concluding sentence in the abstract, page 2, lines 35-36.

Audience:

broad, basic research.

Full Revision

Field of expertise:
cell biology of cytokinesis